# Memory Replay with Data Compression for Continual Learning

**Liyuan Wang**[1,2,3*]   **Xingxing Zhang**[1*]   **Kuo Yang**[6]   **Longhui Yu**[6]   **Chongxuan Li**[4,5†]

**Lanqing Hong**[6]   **Shifeng Zhang**[6]   **Zhenguo Li**[6]   **Yi Zhong**[2,3†]   **Jun Zhu**[1†]

[1]Dept. of Comp. Sci. & Tech., Institute for AI, BNRist Center, THBI Lab, Tsinghua University
[2]School of Life Sciences, IDG/McGovern Institute for Brain Research, Tsinghua University
[3]Tsinghua-Peking Center for Life Sciences [4]Gaoling School of AI, Renmin University of China
[5]Beijing Key Laboratory of Big Data Management and Analysis Methods [6]Huawei Noah's Ark Lab

## Abstract

Continual learning needs to overcome catastrophic forgetting of the past. Memory replay of representative old training samples has been shown as an effective solution, and achieves the state-of-the-art (SOTA) performance. However, existing work is mainly built on a small memory buffer containing a few original data, which cannot fully characterize the old data distribution. In this work, we propose memory replay with data compression (MRDC) to reduce the storage cost of old training samples and thus increase their amount that can be stored in the memory buffer. Observing that the trade-off between the quality and quantity of compressed data is highly nontrivial for the efficacy of memory replay, we propose a novel method based on determinantal point processes (DPPs) to efficiently determine an appropriate compression quality for currently-arrived training samples. In this way, using a naive data compression algorithm with a properly selected quality can largely boost recent strong baselines by saving more compressed data in a limited storage space. We extensively validate this across several benchmarks of class-incremental learning and in a realistic scenario of object detection for autonomous driving.

## 1 Introduction

The ability to continually learn numerous tasks and infer them together is critical for deep neural networks (DNNs), which needs to mitigate *catastrophic forgetting* (McCloskey & Cohen, 1989) of the past. Memory replay of representative old training samples (referred to as *memory replay*) has been shown as an effective solution, and achieves the state-of-the-art (SOTA) performance (Hou et al., 2019). Existing memory replay approaches are mainly built on a small memory buffer containing a few original data, and try to construct and exploit it more effectively. However, due to the low storage efficiency of saving original data, this strategy of building memory buffer will lose a lot of information about the old data distribution. On the other hand, this usually requires huge extra computation to further mitigate catastrophic forgetting, such as by learning additional parameters (Liu et al., 2021a) or distilling old features (Hu et al., 2021).

Different from "artificial" memory replay in DNNs, a significant feature of biological memory is to encode the old experiences in a highly compressed form and replay them to overcome catastrophic forgetting (McClelland, 2013; Davidson et al., 2009; Carr et al., 2011). Thus the learned information can be maintained in a small storage space as comprehensively as possible, and flexibly retrieved. Inspired by the compression feature of biological memory replay, we propose memory replay with data compression (MRDC), which can largely increase the amount of old training samples that can be stored in the memory buffer by reducing their storage cost in a computationally efficient way.

---

*Equal Contribution
†Corresponding Authors

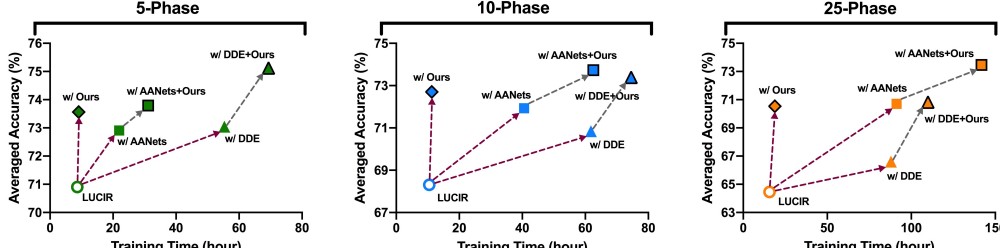

Figure 1: Averaged incremental accuracy and training time on ImageNet-sub. Using JPEG for data compression can achieve comparable or better performance than recent strong approaches with less extra computation (purple arrow), and can further improve their performance (gray arrow).

Given a limited storage space, data compression introduces an additional degree of freedom to explicitly balance the quality and quantity for memory replay. With a properly selected quality, using a naive JPEG compression algorithm (Wallace, 1992) can achieve comparable or better performance than recent strong approaches with less extra computation (Fig. 1, purple arrow), and can further improve their performance (Fig. 1, gray arrow). However, to empirically determine the compression quality is usually inefficient and impractical, since it requires learning a task sequence or sub-sequence repeatedly[1]. We propose a novel method based on determinantal point processes (DPPs) to efficiently determine it without repetitive training. Further, we demonstrate the advantages of our proposal in realistic applications such as continual learning of object detection for autonomous driving, where the incremental data are extremely large-scale.

Our contributions include: (1) We propose memory replay with data compression, which is both an important baseline and a promising direction for continual learning; (2) We empirically validate that the trade-off between quality and quantity of compressed data is highly nontrivial for memory replay, and provide a novel method to efficiently determine it without repetitive training; (3) Extensive experiments show that using a naive data compression algorithm with a properly selected quality can largely improve memory replay by saving more compressed data in a limited storage space.

## 2 RELATED WORK

**Continual learning** needs to overcome catastrophic forgetting of the past when learning a new task. Regularization-based methods (Kirkpatrick et al., 2017; Wang et al., 2021b) approximated the importance of each parameter to the old tasks and selectively penalized its changes. Architecture-based methods (Rusu et al., 2016) allocated a dedicated parameter subspace for each task to prevent mutual interference. Replay-based methods (Rebuffi et al., 2017; Shin et al., 2017) approximated and recovered the old data distribution. In particular, memory replay of representative old training samples (referred to as *memory replay*) can generally achieve the best performance in class-incremental learning (Liu et al., 2021a; Hu et al., 2021) and in numerous other continual learning scenarios, such as audio tasks (Ehret et al., 2020), few-shot (Tao et al., 2020b), semi-supervised (Wang et al., 2021a), and unsupervised continual learning (Khare et al., 2021).

Most of the work in memory replay attempted to more effectively construct and exploit a small memory buffer containing a few original data. As the pioneer work, iCaRL (Rebuffi et al., 2017) proposed a general protocol of memory replay for continual learning. To better construct the memory buffer, Mnemonics (Liu et al., 2020b) parameterized the original data and made them optimizable, while TPCIL (Tao et al., 2020a) constructed an elastic Hebbian graph by competitive Hebbian learning. On the other hand, BiC (Wu et al., 2019), LUCIR (Hou et al., 2019), PODNet (Douillard et al., 2020), DDE (Hu et al., 2021) and AANets (Liu et al., 2021a) attempted to better exploit the memory buffer, such as by mitigating the data imbalance between old and new classes (Hou et al., 2019; Wu et al., 2019; Hu et al., 2021).

In contrast to saving original data, several work attempted to improve the efficiency of remembering the old data distribution. One solution is to continually learn a generative model to replay generated data (Shin et al., 2017; Wu et al., 2018) or compress old training data (Caccia et al., 2020). However,

---

[1]A naive grid search approach is to train continual learning processes with different qualities and choose the best one, resulting in huge computational cost. Also, this strategy will be less applicable if the old data cannot be revisited, or the future data cannot be accessed immediately.

continual learning of such a generative model is extremely challenging, which limits its applications to relatively simple domains, and usually requires a lot of extra computation. Another solution is feature replay: GFR (Liu et al., 2020a) learned a feature generator from a feature extractor to replay generated features, but the feature extractor suffered from catastrophic forgetting since it was incrementally updated. REMIND (Hayes et al., 2020) saved the old features and reconstructed the synthesized features for replay, but it froze the majority of feature extractor after learning the initial phase, limiting the learning of representations for incremental tasks.

**Data compression** aims to improve the storage efficiency of a file, including lossless compression and lossy compression. Lossless compression needs to perfectly reconstruct the original data from the compressed data, which limits its compression rate (Shannon, 1948). In contrast, lossy compression can achieve a much higher compression rate by degrading the original data, so it has been broadly used in realistic applications. Representative hand-crafted approaches include JPEG (or JPG) (Wallace, 1992), which is the most commonly-used algorithm of lossy compression (Mentzer et al., 2020), WebP (Lian & Shilei, 2012) and JPEG2000 (Rabbani, 2002). On the other hand, neural compression approaches generally rely on optimizing Shannon's rate-distortion trade-off, through RNNs (Toderici et al., 2015; 2017), auto-encoders (Agustsson et al., 2017) and GANs (Mentzer et al., 2020).

## 3    CONTINUAL LEARNING PRELIMINARIES

We consider a general setting of continual learning that a deep neural network (DNN) incrementally learns numerous tasks from their task-specific training dataset $D_t = \{(x_{t,i}, y_{t,i})\}_{i=1}^{N_t}$, where $D_t$ is only available when learning task $t$, $(x_{t,i}, y_{t,i})$ is a data-label pair and $N_t$ is the number of such training samples. For classification tasks, the training samples of each task might be from one or several new classes. All the classes ever seen are evaluated at test time, and the classes from different tasks need to be distinguished. This setting is also called class-incremental learning (van de Ven & Tolias, 2019). Suppose such a DNN with parameter $\theta$ has learned $T$ tasks and attempts to learn a new task. Since the old training datasets $\bigcup_{t=1}^{T} D_t$ are unavailable, the DNN will adapt the learned parameters to fit $D_{T+1}$, and tend to catastrophically forget the old tasks McClelland et al. (1995).

An effective solution of overcoming catastrophic forgetting is to select and store representative old training samples $D_t^{mb} = \{(x_{t,i}, y_{t,i})\}_{i=1}^{N_t^{mb}}$ in a small memory buffer ($mb$), and replay them when learning the new task. For classification tasks, mean-of-feature is a widely-used strategy to select $D_t^{mb}$ (Rebuffi et al., 2017; Hou et al., 2019). After learning each task, features of the training data can be obtained by the learned embedding function $F_\theta^e(\cdot)$. In each class, several data points nearest to the mean-of-feature are selected into the memory buffer. Then, the training dataset of task $T + 1$ becomes $D'_{T+1} = D_{T+1} \bigcup D_{1:T}^{mb}$, including both new training samples $D_{T+1}$ and some old training samples $D_{1:T}^{mb} = \bigcup_{t=1}^{T} D_t^{mb}$, so as to prevent forgetting of the old tasks.

However, due to the limited storage space, only a few original data can be saved in the memory buffer, namely, $N_t^{mb} \ll N_t$. Although numerous efforts in memory replay attempted to more effectively exploit the memory buffer, such as by alleviating the data imbalance between the old and new classes, this strategy of building memory buffer is less effective for remembering the old data distribution.

## 4    METHOD

In this section, we first present memory replay with data compression for continual learning. Then, we empirically validate that there is a trade-off between the quality and quantity of compressed data, which is highly nontrivial for memory replay, and propose a novel method to determine it efficiently.

### 4.1    MEMORY REPLAY WITH DATA COMPRESSION

Inspired by the biological memory replay that is in a highly compressed form (Carr et al., 2011), we propose an important baseline for memory replay, that is, using data compression to increase the amount of old training samples that can be stored in the memory buffer, so as to more effectively recover the old data distribution. Data compression can be generally defined as a function $F_q^c(\cdot)$ of compressing the original data $x_{t,i}$ to $x_{q,t,i} = F_q^c(x_{t,i})$ with a controllable quality $q$. Due to the smaller storage cost of each $x_{q,t,i}$ than $x_{t,i}$, the memory buffer can maintain more old training samples for

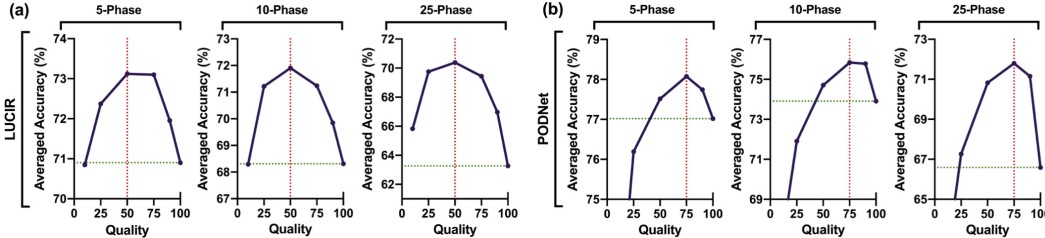

Figure 2: Memory replay with data compression on ImageNet-sub. We make a grid search of the JPEG quality in $\{10, 25, 50, 75, 90\}$. The quality of 100 refers to the original data without compression.

replay, namely, $N_{q,t}^{mb} > N_t^{mb}$ for $D_{q,t}^{mb} = \{(x_{q,t,i}, y_{t,i})\}_{i=1}^{N_{q,t}^{mb}}$ in $D_{q,1:T}^{mb} = \bigcup_{t=1}^{T} D_{q,t}^{mb}$. For notation clarity, we will use $D_q^{mb}$ to denote $D_{q,t}^{mb}$ without explicitly writing out its task label $t$, likewise for $x_i$, $y_i$, $N_q^{mb}$, $N^{mb}$ and $D$. The compression rate can be defined as $r_q = N_q^{mb}/N^{mb} \propto N_q^{mb}$.

In analogy to the learning theory for supervised learning, we argue that continual learning will also benefit from replaying more compressed data, assuming that they approximately follow the original data distribution. However, the assumption is likely to be violated if the compression rate is too high. Intuitively, this leads to a trade-off between quality and quantity: if the storage space is limited, reducing the quality $q$ of data compression will increase the quantity $N_q^{mb}$ of compressed data that can be stored in the memory buffer, and vice versa.

Here we evaluate the proposed idea by compressing images with JPEG (Wallace, 1992), a naive but commonly-used lossy compression algorithm. JPEG can save images with a quality in the range of $[1, 100]$, where reducing the quality results in a smaller file size. Using a memory buffer equivalent to 20 original images per class (Hou et al., 2019), we make a grid search of the JPEG quality with representative memory replay approaches, such as LUCIR (Hou et al., 2019) and PODNet (Douillard et al., 2020). As shown in Fig. 2, memory replay of compressed data with an appropriate quality can substantially outperform that of original data. However, whether the quality is too large or too small, it will affect the performance. In particular, the quality that achieves the best performance varies with the memory replay approaches, but is consistent for different numbers of splits of incremental phases.

## 4.2 QUALITY-QUANTITY TRADE-OFF

Since the trade-off between quality $q$ and quantity $N_q^{mb}$ is highly nontrivial for memory replay, it needs to be carefully determined. Formally, after learning each task from its training dataset $D$, let's consider several compressed subsets $D_q^{mb} = \{(x_{q,i}, y_i)\}_{i=1}^{N_q^{mb}}$, where $q$ is from a set of finite candidates $Q = \{q_1, q_2, q_3, ...\}$. Each $D_q^{mb}$ is constructed by selecting a subset $D_q^{mb*} = \{(x_i, y_i)\}_{i=1}^{N_q^{mb}}$ of $N_q^{mb}$ original training samples from $D$ (following mean-of-feature or other principles). The size $N_q^{mb}$ is determined as the maximum number such that the compressed version of $D_q^{mb*}$ to a quality $q$ can be stored in the memory buffer (see details in Appendix A). Thereby, a smaller $q$ enables to save a larger $N_q^{mb}$, and vice versa. The objective is to select a compressed subset that can best represent $D$, namely, to determine an appropriate $q$ for memory replay.

To understand the effects of data compression, which depends on the compression function $F_q^c(\cdot)$ and the continually-learned embedding function $F_\theta^e(\cdot)$, we focus on analyzing the features of compressed data $f_{q,i} = F_\theta^e(F_q^c(x_i))$. We first calculate the feature matrix $M_q^c = [\bar{f}_{q,1}, \bar{f}_{q,2}, ..., \bar{f}_{q,N_q^{mb}}]$ of each compressed subset $D_q^{mb}$, where each column vector $\bar{f}_{q,i}$ is obtained by normalizing $f_{q,i}$ under $L_2$-norm to keep $||\bar{f}_{q,i}||_2 = 1$. Similarly, we obtain the feature matrix $M_q^* = [\bar{f}_1, \bar{f}_2, ..., \bar{f}_{N_q^{mb}}]$ of each original subset $D_q^{mb*}$. Then, we can analyze the quality-quantity trade-off from two aspects:

On the **empirical** side, in Fig. 3 we use t-SNE (Van der Maaten & Hinton, 2008) to visualize features of the original subset (light dot), which includes different amounts of original data, and its compressed subset (dark dot), which is obtained by compressing the original subset to just fit in the memory buffer. With the increase of quantity and the decrease of quality, the area of compressed subset is initially similar to that of original subset and expands synchronously. However, as a large number of low-quality compressed data occur out-of-distribution, the area of compressed subset becomes much larger than that of its original subset, where the performance also severely declines (see Fig. 2, 3).

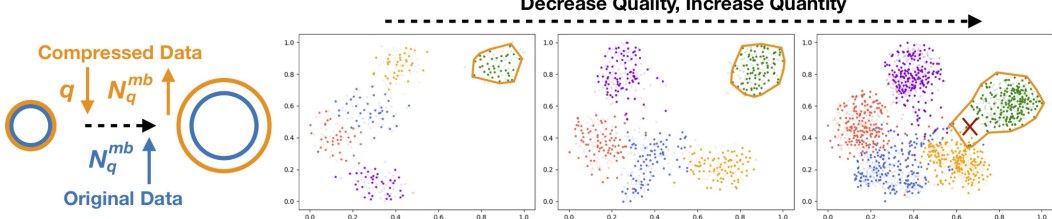

Figure 3: t-SNE visualization of features of the original subset (light dots) and its compressed subset (dark dots) after learning 5-phase ImageNet-sub with LUCIR. From left to right, the quantity is increased from 37, 85 to 200, while the JPEG quality is reduced from 90, 50 to 10. We plot five classes out of the latest task and label them in different colors. The crossed area is out-of-distribution.

On the **theoretical** side, given a training dataset $D$, we aim to find the compressed subset $D_q^{mb}$ that can best represent $D$ by choosing an appropriate compression quality $q$ from its range. To achieve this goal, we introduce $\mathcal{P}_q(D_q^{mb}|D)$ to characterize the conditional likelihood of selecting $D_q^{mb}$ given input $D$ under parameter $q$. The goal of learning is to choose appropriate $q$ based on the training tasks for making accurate predictions on unseen inputs. While there are a variety of objective functions for learning, here we focus on the widely-used maximum likelihood estimation (MLE), where the goal is to choose $q$ to maximize the conditional likelihood of the observed data:

$$\max_q \ \mathcal{P}_q(D_q^{mb}|D). \tag{1}$$

The construction of $D_q^{mb}$ can be essentially viewed as a sampling problem with the cardinality $N_q^{mb}$. Here, we apply Determinantal Point Processes (DPPs) to formulate the conditional likelihood $\mathcal{P}_q(D_q^{mb}|D)$, since DPPs are not only elegant probabilistic sampling models (Kulesza & Taskar, 2012), which can characterize the probabilities for every possible subset by determinants, but also provide a geometric interpretation of the probability by the volume spanned by all elements in the subset (detailed in Appendix C.2). In particular, a conditional DPP is a conditional probabilistic model which assigns a probability $\mathcal{P}_q(D_q^{mb}|D)$ to each possible subset $D_q^{mb}$. Since the network parameter $\theta$ is fixed during compression, and the feature matrix $M_q^c = F_\theta^e(D_q^{mb})$, we rewrite $\mathcal{P}_q(D_q^{mb}|D)$ as $\mathcal{P}_q(M_q^c|D)$ equivalently. Formally, such a DPP formulates the probability $\mathcal{P}_q(D_q^{mb}|D)$ as

$$\mathcal{P}_q(M_q^c|D) = \frac{\det(L_{M_q^c}(D;q,\theta))}{\sum_{|M|=N_q^{mb}} \det(L_M(D;q,\theta))}, \tag{2}$$

where $|M_q^c| = N_q^{mb}$ and $L(D;q,\theta)$ is a conditional DPP $|D| \times |D|$ kernel matrix that depends on the input $D$, the parameters $\theta$ and $q$. $L_M(D;q,\theta)$ (resp., $L_{M_q^c}(D;q,\theta)$) is the submatrix sampled from $L(D;q,\theta)$ using indices from $M$ (resp., $M_q^c$). The numerator defines the marginal probability of inclusion for the subset $M_q^c$, and the denominator serves as a normalizer to enforce the sum of $\mathcal{P}_q(M_q^c|D)$ for every possible $M_q^c$ to 1. Generally, there are many ways to obtain a positive semi-definite kernel $L$. In this work, we employ the most widely-used dot product kernel function, where $L_{M_q^c} = M_q^{c\top} M_q^c$ and $L_M = M^\top M$.

However, due to the extremely high complexity of calculating the denominator in Eq. (2) (analyzed in Appendix C.1), it is difficult to optimize $\mathcal{P}_q(M_q^c|D)$. Alternatively, by introducing $\mathcal{P}_q(M_q^*|D)$ to characterize the conditional likelihood of selecting $M_q^*$ given input $D$ under parameter $q$, we propose a relaxed optimization program of Eq. (1), in which we (1) maximize $\mathcal{P}_q(M_q^*|D)$ since $\mathcal{P}_q(M_q^c|D) \leq \mathcal{P}_q(M_q^*|D)$ is always satisfied under lossy compression; and meanwhile, (2) constrain that $\mathcal{P}_q(M_q^c|D)$ is consistent with $\mathcal{P}_q(M_q^*|D)$. The relaxed program is solved as follows.

First, by formulating $\mathcal{P}_q(M_q^*|D)$ similarly as $\mathcal{P}_q(M_q^c|D)$ in Eq. (2) (detailed in Appendix C.2), we need to maximize

$$\mathcal{L}_1(q) = \mathcal{P}_q(M_q^*|D) = \frac{\det(L_{M_q^*}(D;\theta))}{\sum_{|M^*|=N_q^{mb}} \det(L_{M^*}(D;\theta))}, \tag{3}$$

where the conditional DPP kernel matrix $L(D;\theta)$ only depends on $D$ and $\theta$. For our task, $\mathcal{P}_q(M_q^*|D)$ monotonically increases with $N_q^{mb}$. Thus, optimizing $\mathcal{L}_1$ is converted into $\max_q \ N_q^{mb}$ equivalently, with significantly reduced complexity (detailed in Proposition 1 of Appendix C.3).

Second, to constrain that $\mathcal{P}_q(M_q^c|D)$ is consistent with $\mathcal{P}_q(M_q^*|D)$, we propose to minimize

$$\mathcal{L}_2(q) = \left| \frac{\mathcal{P}_q(M_q^c|D)}{\mathcal{P}_q(M_q^*|D)} - 1 \right| = \left| \frac{\det(M_q^{c\top} M_q^c)}{\det(M_q^{*\top} M_q^*)} Z_q - 1 \right| = \left| \left( \frac{\text{Vol}_q^c}{\text{Vol}_q^*} \right)^2 Z_q - 1 \right| = \left| R_q^2 Z_q - 1 \right|, \quad (4)$$

where $Z_q = \frac{\sum_{|M^*|=N_q^{mb}} \det(L_{M^*}(D;\theta))}{\sum_{|M|=N_q^{mb}} \det(L_M(D;q,\theta))}$. In particular, $\det(M_q^{*\top} M_q^*)$ has a geometric interpretation that it is equal to the squared volume spanned by $M_q^*$ (Kulesza & Taskar, 2012), denoted as $\text{Vol}_q^*$, likewise for $\det(M_q^{c\top} M_q^c)$ with respect to $\text{Vol}_q^c$. Then we define $R_q = \frac{\text{Vol}_q^c}{\text{Vol}_q^*}$ as the ratio of the two feature volumes. To avoid computing $Z_q$, we can convert optimizing $\mathcal{L}_2$ into minimizing $|R_q - 1|$ equivalently, since both of them mean maximizing $q$ (detailed in Proposition 2 of Appendix C.4).

Putting $\mathcal{L}_1$ and $\mathcal{L}_2$ together, our method is finally reformulated as

$$\max_q \ g(q)$$
$$s.t., \ q \in Q, \ |R_q - 1| < \epsilon, \quad (5)$$

where $\epsilon$ is a small positive number to serve as the threshold of $R_q$. $g(\cdot) : \mathbb{R} \to \mathbb{N}$ represents the function that outputs the maximum number (i.e., $N_q^{mb}$) such that the compressed version of $D_q^{mb*}$ to a quality $q$ can be stored in the memory buffer. Note that we relax the original restriction of minimizing $|R_q - 1|$ by enforcing $|R_q - 1| < \epsilon$, since maximizing $N_q^{mb}$ for $\mathcal{L}_1$ and maximizing $q$ for $\mathcal{L}_2$ cannot be achieved simultaneously. Of note, $R_q$ is calculated by normalizing the feature volume $\text{Vol}_q^c$ with $\text{Vol}_q^*$, both of which depend on $q$ (i.e., $N_q^{mb}$). Therefore, $\epsilon$ can largely mitigate the sensitivity of $q$ to various domains and can be empirically set as a constant value (see Sec. 4.3).

Since the function $g(\cdot)$ in Eq. (5) is highly non-smooth, gradient-based methods are not applicable. Indeed, we solve it by selecting the best candidate in a finite-size set $Q$. Generally, the candidate values in $Q$ can be equidistantly selected from the range of $q$, such as $[1, 100]$ for JPEG. More candidate values can determine a proper $q$ more accurately, but the complexity will grow linearly. We found that selecting 5 candidate values is a good choice in our experiments. Once we solve Eq. (5), a good trade-off is achieved by reducing $q$ as much as possible to obtain a larger $N_q^{mb}$, while keeping the feature volume $\text{Vol}_q^c$ similar to $\text{Vol}_q^*$. This is consistent with our empirical analysis in Fig. 3.

### 4.3 VALIDATE OUR METHOD WITH GRID SEARCH RESULTS

In essence, the grid search described in Sec. 4.1 can be seen as a naive approach to determine the compression quality, which is similar to selecting other hyperparameters for continual learning (Fig. 4, a). This strategy is to learn a task sequence or sub-sequence using different qualities and choose the best one (Fig. 4, b), which leads to huge extra computation and will be less applicable if the old data cannot be revisited, or the future data cannot be accessed immediately. In contrast, our method described in Sec. 4.2 only needs to calculate the feature volumes of each compressed subset $D_q^{mb}$ and original subset $D_q^{mb*}$ (Fig. 4, c), without repetitive training.

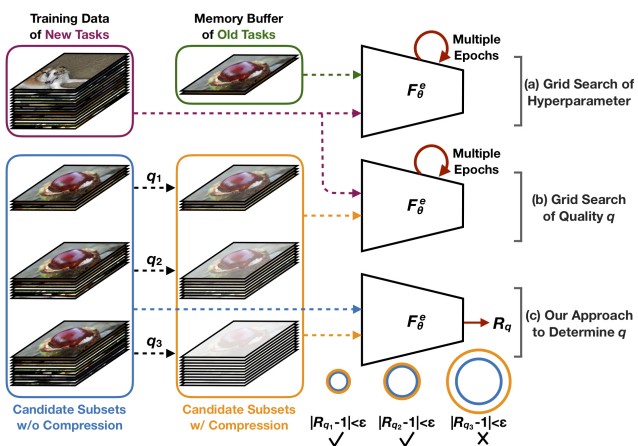

Figure 4: Properly select a quality without repetitive training.

Now we validate the quality determined by our method with the grid search results, where LUCIR and PODNet achieve the best performance at the JPEG quality of 50 and 75 on ImageNet-sub, respectively (see Fig. 2). We present $R_q$ in each incremental phase and the averaged $R_q$ of all incremental phases for 5-phase split in Fig. 5 and for 10- and 25-phase splits in Appendix Fig.13. Based on the principle in Eq. (5) with $\epsilon = 0.5$, it can be clearly seen that 50 and 75 are the qualities chosen for LUCIR and

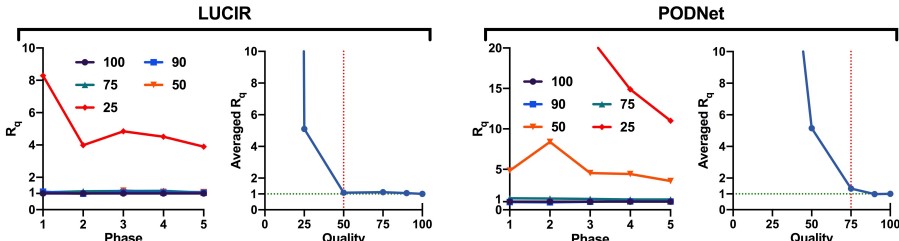

Figure 5: For 5-phase ImageNet-sub, we present $R_q$ in each incremental phase with various compression qualities $q$, and the averaged $R_q$ of all incremental phases.

PODNet, respectively, since they are the smallest qualities that satisfy $|R_q - 1| < \epsilon$. Therefore, the quality determined by our method is consistent with the grid search results, but the computational cost is saved by more than 100 times. Interestingly, for each quality $q$, whether $|R_q - 1| < \epsilon$ is generally consistent in each incremental phase and the average of all incremental phases. We further explore the scenarios where $R_q$ might be more dynamic in Appendix D.4.

## 5 EXPERIMENT

In this section, we first evaluate memory replay with data compression (MRDC) in class-incremental learning of large-scale images. Then, we demonstrate the advantages of our proposal in realistic semi-supervised continual learning of large-scale object detection for autonomous driving.[2]

### 5.1 CLASS-INCREMENTAL LEARNING

**Benchmark**: We consider three benchmark datasets of large-scale images for continual learning. CUB-200-2011 (Wah et al., 2011) is a large-scale dataset including 200-class 11,788 colored images of birds with default format of JPG, split as 30 images per class for training while the rest for testing. ImageNet-full (Russakovsky et al., 2015) includes 1000-class large-scale natural images with default format of JPEG. ImageNet-sub (Hou et al., 2019) is a subset derived from ImageNet-full, consisting of randomly selected 100 classes of images. Following Hou et al. (2019), we randomly resize, crop and normalize the images to the size of $224 \times 224$, and randomly split a half of the classes as the initial phase while split the rest into 5, 10 and 25 incremental phases. We report the averaged incremental accuracy with single-head evaluation (Chaudhry et al., 2018) in the main text, and further present the averaged forgetting in Appendix E.3.

**Implementation**: We follow the implementation of representative memory replay approaches (Hou et al., 2019; Douillard et al., 2020) (detailed in Appendix B.1), where we focus on constraining a certain storage space of the memory buffer rather than a certain number of images. The storage space is limited to the equivalent of 20 original images per class, if not specified. We further discuss the effects of different storage space in Fig. 7, a fixed memory budget in Appendix E.4 and less compressed samples in Appendix E.5. For data compression, we apply a naive but commonly-used JPEG (Wallace, 1992) algorithm to compress images to a controllable quality in the range of $[1, 100]$.

**Baseline**: We evaluate representative memory replay approaches such as LwF (Li & Hoiem, 2017), iCaRL (Rebuffi et al., 2017), BiC (Wu et al., 2019), LUCIR (Hou et al., 2019), Mnemonics (Liu et al., 2020b), TPCIL (Tao et al., 2020a), PODNet (Douillard et al., 2020), DDE (Hu et al., 2021) and AANets (Liu et al., 2021a). In particular, AANets and DDE are the *recent strong approaches* implemented on the backbones of LUCIR and PODNet, so we also implement ours on the two backbones. Since both AANets and DDE only release their official implementation on LUCIR, we further reproduce LUCIR w/ AANets and LUCIR w/ DDE for fair comparison.

**Accuracy:** We summarize the performance of the above baselines and memory replay with data compression (MRDC, ours) in Table 1. Using the same extra storage space, ours achieves comparable or better performance than AANets and DDE on the same backbone approaches, and can further boost their performance by a large margin. The improvement from ours is due to mitigating the averaged forgetting, detailed in Appendix E.3. For CUB-200-2011 with only a few training samples, all old data can be saved in the memory buffer with a high JPEG quality of 94. In contrast, for ImageNet-sub/-full, only a part of old training samples can be selected, compressed and saved in the

---

[2] All experiments are averaged by more than three runs with different random seeds.

Table 1: Averaged incremental accuracy (%) of classification tasks. The reproduced results are presented with $\pm$ standard deviation, while others are reported results. The reproduced results might slightly vary from the reported results due to different random seeds. [1]With class-balance finetuning. [2]PODNet reproduced by Hu et al. (2021) underperforms that in Douillard et al. (2020).

| Method | CUB-200-2011 | | | ImageNet-sub | | | ImageNet-full | |
|---|---|---|---|---|---|---|---|---|
| | 5-phase | 10-phase | 25-phase | 5-phase | 10-phase | 25-phase | 5-phase | 10-phase |
| LwF (Li & Hoiem, 2017) | 39.42 $\pm 0.48$ | 38.53 $\pm 0.96$ | 36.33 $\pm 0.74$ | 53.62 | 47.64 | 44.32 | 44.35 | 38.90 |
| iCaRL (Rebuffi et al., 2017) | 39.49 $\pm 0.58$ | 39.31 $\pm 0.66$ | 38.77 $\pm 0.73$ | 65.44 | 59.88 | 52.97 | 51.50 | 46.89 |
| BiC (Wu et al., 2019) | 45.29 $\pm 0.88$ | 45.25 $\pm 0.70$ | 45.17 $\pm 0.27$ | 70.07 | 64.96 | 57.73 | 62.65 | 58.72 |
| Mnemonics (Liu et al., 2020b) | – | – | – | 72.58 | 71.37 | 69.74 | 64.54 | 63.01 |
| TPCIL (Tao et al., 2020a) | – | – | – | 76.27 | 74.81 | – | 64.89 | 62.88 |
| LUCIR (Hou et al., 2019) | 44.63 $\pm 0.32$ | 45.58 $\pm 0.28$ | 45.48 $\pm 0.66$ | 70.84 | 68.32 | 61.44 | 64.45 | 61.57 |
| w/ AANets (Liu et al., 2021a) | – | – | – | 72.55 | 69.22 | 67.60 | 64.94 | 62.39 |
| w/ DDE (Hu et al., 2021) | – | – | – | 72.34 | 70.20 | – | 67.51[1] | 65.77[1] |
| *w/ MRDC (Ours)* | 46.68 $\pm 0.60$ | 47.28 $\pm 0.51$ | 48.01 $\pm 0.72$ | 73.56 $\pm 0.27$ | 72.70 $\pm 0.47$ | 70.53 $\pm 0.57$ | 67.53 $\pm 0.08$[1] | 65.29 $\pm 0.10$[1] |
| w/ AANets (Reproduced) | 46.87 $\pm 0.66$ | 47.34 $\pm 0.77$ | 47.35 $\pm 0.95$ | 72.91 $\pm 0.45$ | 71.93 $\pm 0.52$ | 70.70 $\pm 0.46$ | 63.37 $\pm 0.26$ | 62.46 $\pm 0.14$ |
| *w/ AANets + MRDC (Ours)* | **49.02** $\pm 1.07$ | **49.84** $\pm 0.87$ | **51.33** $\pm 1.42$ | 73.79 $\pm 0.42$ | 73.73 $\pm 0.37$ | **73.47** $\pm 0.35$ | 64.99 $\pm 0.13$ | 63.04 $\pm 0.11$ |
| w/ DDE (Reproduced) | 45.86 $\pm 0.65$ | 46.48 $\pm 0.69$ | 46.46 $\pm 0.33$ | 73.04 $\pm 0.36$ | 70.84 $\pm 0.59$ | 66.61 $\pm 0.68$ | 66.95 $\pm 0.09$[1] | 65.21 $\pm 0.05$[1] |
| *w/ DDE + MRDC (Ours)* | 47.16 $\pm 0.60$ | 48.33 $\pm 0.48$ | 48.37 $\pm 0.34$ | 75.12 $\pm 0.17$ | 73.39 $\pm 0.29$ | 70.83 $\pm 0.34$ | 67.90 $\pm 0.05$[1] | **66.67** $\pm 0.23$[1] |
| PODNet (Douillard et al., 2020) | 44.92 $\pm 0.31$ | 44.49 $\pm 0.65$ | 43.79 $\pm 0.44$ | 75.54 | 74.33 | 68.31 | 66.95 | 64.13 |
| w/ AANets (Liu et al., 2021a) | – | – | – | 76.96 | 75.58 | 71.78 | 67.73 | 64.85 |
| w/ DDE (Hu et al., 2021) | – | – | – | 76.71 | 75.41 | – | 66.42[2] | 64.71 |
| *w/ MRDC (Ours)* | 46.00 $\pm 0.28$ | 46.09 $\pm 0.37$ | 45.84 $\pm 0.43$ | **78.08** $\pm 0.66$ | **76.02** $\pm 0.54$ | 72.72 $\pm 0.74$ | **68.91** $\pm 0.16$ | 66.31 $\pm 0.26$ |

memory buffer, where our method can efficiently determine the compression quality for LUCIR and PODNet (see Sec. 4.3 and Appendix D.2), and for AANets and DDE (see Appendix D.3).

**Computational Cost:** Limiting the size of memory buffer is not only to save its storage, but also to save the extra computation of learning all old training samples again. Here we evaluate the computational cost of AANets, DDE and memory replay with data compression (MRDC, ours) on LUCIR (see Fig. 6 for ImageNet-sub and Appendix Table 3 for CUB-200-2011). Both AANets and DDE require a huge amount of extra computation to improve the performance of continual learning, which is gener-

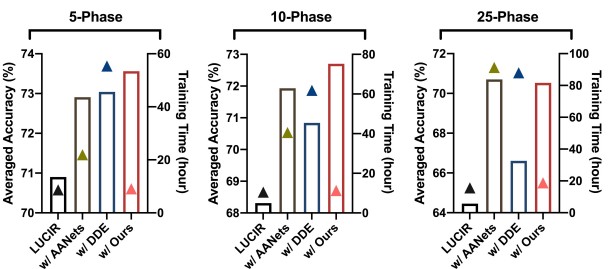

Figure 6: Averaged incremental accuracy (the column, left Y-axis) and computational cost (the triangle, right Y-axis) on ImageNet-sub. We run each baseline with one Tesla V100.

ally *several times* that of the backbone approach. In contrast, ours achieves competing or more performance improvement but only slightly increases the computational cost.

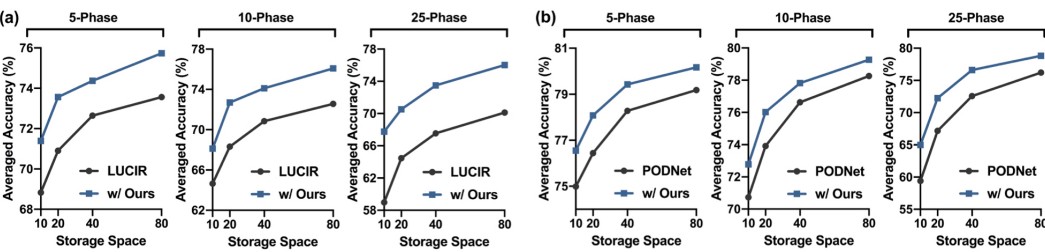

Figure 7: The effects of different storage space (equal to 10, 20, 40 and 80 original images per class) on ImageNet-sub. LUCIR (a) and PODNet (b) are reproduced from their officially-released codes.

**Storage Space:** The impact of storage space is evaluated in Fig. 7. Limiting the storage space to equivalent of 10, 20, 40 and 80 original images per class, ours can improve the performance of LUCIR and PODNet by a similar margin, where the improvement is generally more significant for

more splits of incremental phases. Further, ours can achieve consistent performance improvement when using a fixed memory budget (detailed in Appendix E.4), and can largely save the storage cost without performance decrease when storing less compressed samples (detailed in Appendix E.5).

## 5.2 LARGE-SCALE OBJECT DETECTION

The advantages of memory replay with data compression are more significant in realistic scenarios such as autonomous driving, where the incremental data are extremely large-scale with huge storage cost. Here we evaluate continual learning on SODA10M (Han et al., 2021), a large-scale object detection benchmark for autonomous driving. SODA10M contains 10M unlabeled images and 20K labeled images of size $1920 \times 1080$ with default format of JPEG, which is much larger than the scale of ImageNet. The labeled images are split into 5K, 5K and 10K for training, validation and testing, respectively, annotating 6 classes of road objects for detection.

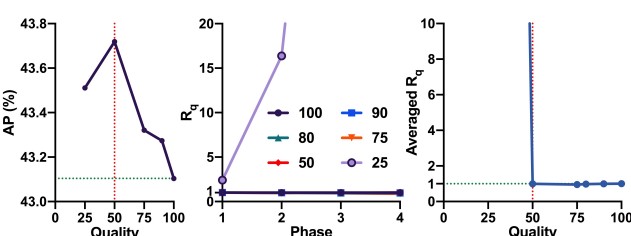

Figure 8: We present the grid search results (left), $R_q$ in each incremental phase (middle), and the averaged $R_q$ of all incremental phases (right) for SSCL on SODA10M. The quality determined by our method is 50, which indeed achieves the best performance in the grid search.

Since the autonomous driving data are typically partially-labeled, we follow the semi-supervised continual learning (SSCL) proposed by Wang et al. (2021a) for object detection. Specifically, we randomly split 5 incremental phases containing 1K labeled data and 20K unlabeled data per phase, and use a memory buffer to replay labeled data. The storage space is limited to the equivalent of 100 original images per phase. Following Han et al. (2021), we consider Pseudo Labeling

Table 2: Detection results with $\pm$ standard deviation (%) of semi-supervised continual learning on SODA10M. FT: finetuning. MR: memory replay.

|  | Method | AP | $AP_{50}$ | $AP_{75}$ |
|---|---|---|---|---|
| Pseudo Labeling | FT | $40.36_{\pm 0.34}$ | $63.83_{\pm 0.35}$ | $43.82_{\pm 0.34}$ |
|  | MR | $40.75_{\pm 0.30}$ / +0.39 | $65.11_{\pm 0.64}$ / +1.28 | $43.53_{\pm 0.20}$ / -0.29 |
|  | Ours | $\mathbf{41.50}_{\pm 0.06}$ / **+1.14** | $\mathbf{65.36}_{\pm 0.47}$ / **+1.53** | $\mathbf{44.95}_{\pm 0.22}$ / **+1.13** |
| Unbiased Teacher | FT | $42.88_{\pm 0.32}$ | $66.70_{\pm 0.59}$ | $45.99_{\pm 0.32}$ |
|  | MR | $43.10_{\pm 0.06}$ / +0.22 | $66.88_{\pm 0.51}$ / +0.18 | $46.62_{\pm 0.02}$ / +0.63 |
|  | Ours | $\mathbf{43.72}_{\pm 0.25}$ / **+0.84** | $\mathbf{67.80}_{\pm 0.46}$ / **+1.10** | $\mathbf{47.36}_{\pm 0.23}$ / **+1.37** |

and Unbiased Teacher (Liu et al., 2021b) for semi-supervised object detection (the implementation is detailed in Appendix B.2). Using the method described in Sec. 4.2 with the same threshold (i.e., $\epsilon = 0.5$), we can efficiently determine the compression quality for SSCL of object detection, and validate that the determined quality indeed achieves the best performance in grid search (see Fig. 8). Then, compared with memory replay of original data, our proposal can generally achieve *several times* of the performance improvement on finetuning, as shown in Table 2.

## 6 CONCLUSION

In this work, we propose that using data compression with a properly selected compression quality can largely improve the efficacy of memory replay by saving more compressed data in a limited storage space. To efficiently determine the compression quality, we provide a novel method based on determinantal point processes (DPPs) to avoid repetitive training, and validate our method in both class-incremental learning and semi-supervised continual learning of object detection. Our work not only provides an important yet under-explored baseline, but also opens up a promising new avenue for continual learning. Further work could develop adaptive compression algorithms for incremental data to improve the compression rate, or propose new regularization methods to constrain the distribution changes caused by data compression. Meanwhile, the theoretical analysis based on DPPs can be used as a general framework to integrate optimizable variables in memory replay, such as the strategy of selecting prototypes. In addition, our work suggests how to save a batch of training data in a limited storage space to best describe its distribution, which will motivate broader applications in the fields of data compression and data selection.

ACKNOWLEDGEMENTS

This work was supported by NSF of China Projects (Nos. 62061136001, 61620106010, 62076145, U19B2034, U181146), Beijing NSF Project (No. JQ19016), Beijing Outstanding Young Scientist Program NO. BJJWZYJH012019100020098, Tsinghua-Peking Center for Life Sciences, Beijing Academy of Artificial Intelligence (BAAI), Tsinghua-Huawei Joint Research Program, a grant from Tsinghua Institute for Guo Qiang, and the NVIDIA NVAIL Program with GPU/DGX Acceleration, Major Innovation & Planning Interdisciplinary Platform for the "Double-First Class" Initiative, Renmin University of China.

ETHIC STATEMENT

This work presents memory replay with data compression for continual learning. It can be used to reduce the storage requirement in memory replay and thus may facilitate large-scale applications of such methods to real-world problems (e.g., autonomous driving). As a fundamental research in machine learning, the negative consequences in the current stage are not obvious.

REPRODUCIBILITY STATEMENT

We ensure the reproducibility of our paper from three aspects. (1) Experiment: The implementation of our experiment described in Sec. 4.1, Sec. 4.3 and Sec. 5 is further detailed in Appendix B. (2) Code: Our code is included in supplementary materials. (3) Theory and Method: A complete proof of the theoretical results described in Sec. 4.2 is included in Appendix C.

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

## A  TRADE-OFF BETWEEN QUALITY AND QUANTITY

Given a limited storage space, there is an intuitive trade-off between the quality $q$ and quantity $N_q^{mb}$ of compressed data. Here we plot the relation of $q$ and $N_q^{mb}$ for ImageNet and SODA10M used in our paper. As shown in Fig. 9, reducing the quality enables to save more compressed data in the memory buffer, and vice versa. Then, we present compressed images with various qualities in Fig. 10. Although reducing the quality tends to distort the compressed data and thus hurts the performance of continual learning, it can be clearly seen that the semantic information of such compressed data is hardly affected. There-

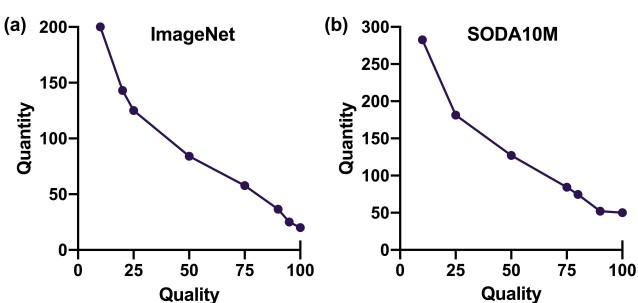

Figure 9: Limiting the storage space to the equivalent of (a) 20 original images per class for ImageNet and (b) 50 original images per phase for SODA10M, we plot the relation between the quality and quantity of compressed data. The quality of 100 refers to the original data without compression.

fore, a potential follow-up work is to regularize the differences between compressed data and original data, so as to further improve the performance of memory replay.

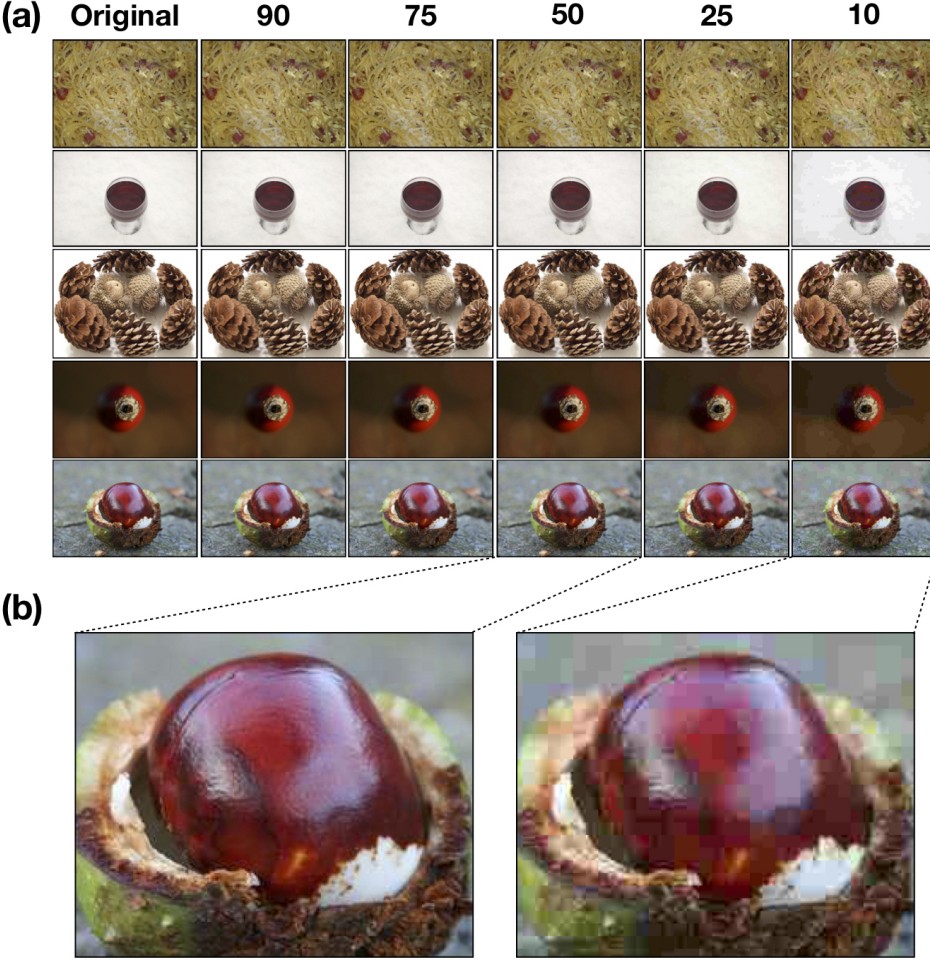

Figure 10: Compressed images of ImageNet. (a) Each column contains compressed images of a specific JPEG quality. (b) Two exemplar images with different qualities.

# B  IMPLEMENTATION DETAIL

## B.1  CLASS-INCREMENTAL LEARNING

Following the implementation of Hou et al. (2019); Douillard et al. (2020); Tao et al. (2020a); Liu et al. (2021a); Hu et al. (2021), we train a ResNet-18 architecture for 90 epochs, with minibatch size of 128 and weight decay of $1 \times 10^{-4}$. We use a SGD optimizer with initial learning rate of 0.1, momentum of 0.9 and cosine annealing scheduling. We run each baseline with one Tesla V100, and measure the computational cost. We evaluate the performance of class-incremental learning with the averaged incremental accuracy (AIC) (Hou et al., 2019). After learning each phase, we can calculate the test accuracy on all classes learned so far in the previous $t$ phases as $A_t$. Then we can calculate $AIC_t = \frac{1}{t} \sum_{i=1}^{t} A_i$.

To evaluate the effects of class similarity on the determined quality of data compression, we select similar or dissimilar superclasses from ImageNet, which includes 1000 classes ranked by their semantic similarity. Specifically, we select 10 adjacent classes as a superclass, and construct 10 adjacent (similar) or remote (dissimilar) superclasses. The index of classes selected in 10 similar superclasses is [1, 2, 3, ..., 100], while in 10 dissimilar superclasses is [1, 2, ..., 10, 101, 102, ..., 110, ..., 910].

## B.2  LARGE-SCALE OBJECT DETECTION

For memory replay, we follow Wang et al. (2021a) to randomly select the old labeled data and save them in the memory buffer, which indeed achieves competitive performance as analyzed by Chaudhry et al. (2018). We follow the implementation of Han et al. (2021) for semi-supervised object detection. Specifically, we use Faster-RCNN (Ren et al., 2015) with FPN (Lin et al., 2017) and ResNet-50 backbone as the object detection network for Pseudo Labeling (Han et al., 2021) and Unbiased Teacher (Liu et al., 2021b). For each incremental phase, we train the network for 10K iterations using the SGD optimizer with initial learning rate of 0.01, momentum of 0.9, and constant learning rate scheduler. The batch size of supervised and unsupervised data are both 16 images. For Pseudo Labeling, we first train a supervised model on the labeled set in the first 2K iterations. Then we evaluate the supervised model on the unlabeled set. A bounding box with a predicted score larger than 0.7 in the evaluation results is selected as a pseudo label, which would be used to train the supervised model in the later 8K iterations in each phase. For Unbiased Teacher, we follow the default setting and change the input size to comply with SODA10M. We run each baseline with 8 Tesla V100. The performance is evaluated by the metrics ($AP$, $AP_{50}$, $AP_{75}$) used in the prior works (Liu et al., 2021b; Han et al., 2021).

# C  THEORETICAL ANALYSIS

## C.1  DETERMINANTAL POINT PROCESSES (DPPS) PRELIMINARIES

Arising in quantum physics and random matrix theory, determinantal point processes (DPPs) are elegant probabilistic models of global, negative correlations, and offer efficient algorithms for sampling, marginalization, conditioning, and other inference tasks.

A point process $\mathcal{P}$ on a ground set $\mathcal{V}$ is a probability measure on the power set $2^N$, where $N = |\mathcal{V}|$ is the size of the ground set. Let $B$ be a $D \times N$ matrix with $D \geq N$. In practice, the columns of $B$ are vectors representing items in the set $\mathcal{V}$. Denote the columns of $B$ by $B_i$ for $i = 1, 2, \cdots, N$. A sample from $\mathcal{P}$ might be the empty set, the entirety of $B$, or anything in between. $\mathcal{P}$ is called a determinantal point process if, given a random subset $Y$ drawn according to $\mathcal{P}$ ($2^N$ possible instantiations for $Y$), we have for every $S \subseteq B$,

$$\mathcal{P}(Y = S) \propto \det(L_S), \tag{6}$$

where $S = \{B_{i_1}, B_{i_2}, \cdots, B_{i_{|S|}}\}$, $\mathcal{P}(Y = S)$ characterizes the likelihood of selecting the subset $S$ from $B$. For some symmetric similarity kernel $L \in \mathbb{R}^{N \times N}$ (e.g., $L = B^{\mathrm{T}} B$), where $L_S$ is the similarity kernel of subset $S$ (e.g., $L = S^{\mathrm{T}} S$). That is, $L_S$ is the submatrix sampled from $L$ using indices from $S$. Since $\mathcal{P}$ is a probability measure, all principal minors $\det(L_S)$ of $L$ must

be non-negative, and thus $L$ itself must be positive semidefinite. These requirements turn out to be sufficient: any $L$, $0 \preceq L \preceq I$, defines a DPP.

$L$ is often referred to as the marginal kernel since it contains all the information needed to compute the probability of any subset $S$ being included in $Y$. Hence, the marginal probability of including one element $B_i$ is $\mathcal{P}(B_i \in Y) = L_{ii}$, and two elements $B_i$ and $B_j$ is $\mathcal{P}(B_i, B_j \in Y) = L_{ii}L_{jj} - L_{ij}^2 = \mathcal{P}(B_i \in Y)\mathcal{P}(B_j \in Y) - L_{ij}^2$. A large value of $L_{ij}$ reduces the likelihood of both elements to appear together in a diverse subset. This demonstrates why DPPs are "diversifying". Below is several important conclusions about DPPs related to our work (Kulesza & Taskar, 2012).

**Geometric Interpretation.** By focusing on L-ensembles in DPPs, determinants have an intuitive geometric interpretation. If $L = B^\top B$, then

$$\mathcal{P}(Y = S) \propto \det(L_S) = (\text{Vol}(S))^2, \tag{7}$$

where the first two terms specify the marginal probabilities for every possible subset $S$, and the last term is the squared $|S|$-dimensional volume of the parallelepiped spanned by the items in $S$.

**Quality vs. Diversity.** While the entries of the DPP kernel $L$ are not totally opaque in that they can be seen as measures of similarity— reflecting our primary qualitative characterization of DPPs as diversifying processes. In most practical situations, we want diversity to be balanced against some underlying preferences for different items in $B$. Kulesza & Taskar (2012) proposed a decomposition of the DPP that more directly illustrates the tension between diversity and a per-item measure of quality. Specifically, $L = B^\top B$ can be taken one step further, by writing each column $B_i$ as the product of a quality term $q_i \geq 0$ and vector of normalized diversity features $f_i \in \mathbb{R}^D$, $\|f_i\| = 1$. Then, $L_{ij} = q_i f_i^\top f_j q_j$, where $f_i^\top f_j \in [-1, 1]$ is a signed measure of similarity between $B_i$ and $B_j$. Let $\phi_{ij} = f_i^\top f_j = \frac{L_{ij}}{\sqrt{L_{ii}L_{jj}}}$, and $\phi = \{\phi_{ij}\}_{i,j=1}^N$. Then, depending on Eq. (6), we have

$$\mathcal{P}(Y = S) \propto \left( \prod_{B_{i_j} \in S} q_{i_j}^2 \right) \det(\phi_S), \tag{8}$$

where $q_{i_j}$ can be seen as a quality score of an item $B_{i_j}$ in $B$, and $\phi_S$ is the submatrix sampled from $\phi$ using indices from $S$.

**Alternative Likelihood Formulas.** In an L-ensemble DPP, the likelihood of a particular set $S \subseteq B$ is given by

$$\mathcal{P}_L(S) = \frac{\det(L_S)}{\sum_{S'} \det(L_{S'})} = \frac{\det(L_S)}{\det(L + I)}, \tag{9}$$

where $S'$ is one of all $2^N$ possible subsets in $B$, and $L_{S'}$ is the submatrix sampled from $L$ using indices from $S'$.

This expression has some nice intuitive properties in terms of volumes, and, ignoring the normalization in the denominator, takes a simple and concise form. However, as a ratio of determinants on matrices of differing dimension, it may not always be analytically convenient. Minors can be difficult to reason about directly, and ratios complicate calculations like derivatives. Such a high computational cost is just one key challenge in DPPs.

**Conditional DPPs.** A conditional DPP $\mathcal{P}(Y = S|B)$ is a conditional probabilistic model which assigns a probability to every possible subset $S \subseteq B$. The model takes the form of an L-ensemble:

$$\mathcal{P}(Y = S|B) \propto \det(L_S(B)), \tag{10}$$

where $L(B)$ is a positive semidefinite conditional $N \times N$ kernel matrix that depends on the input $B$.

**$k$-DPPs.** A $k$-DPP is obtained by conditioning a standard DPP on the event that the set $Y$ has cardinality $k$, which is concerned only with the content of a random $k$-set (i.e., without the size $k$) (Kulesza & Taskar, 2012). Formally, the $k$-DPP $\mathcal{P}_L^k$ gives the probability of a particular set $S$ as

$$\mathcal{P}_L^k(Y = S) = \frac{\det(L_S)}{\sum_{|S'|=k} \det(L_{S'})}, \tag{11}$$

where $|S| = k$ and $L$ is a positive semidefinite kernel. Although numerous optimization algorithms have been proposed to solve DPPs problems, the high computational complexity about denominator cannot be avoided generally. On the one hand, computing denominator is a sum over $\binom{N}{k}$ terms. On the other hand, computing the determinant of each term through matrix decomposition is with $O(k^3)$ time. Just as Kulesza & Taskar (2012) claimed, sparse storage of larger matrices is possible for DPPs, but computing determinants remains prohibitively expensive unless the level of sparsity is extreme.

If the dot product kernel function is adopted to compute the kernel $L$ in Eq. (11), similar to Geometric Interpretation of standard DPPs above, the probability $\mathcal{P}_L^k(Y = S)$ is further defined with volume sampling (Deshpande & Rademacher, 2010), i.e.,

$$\mathcal{P}_L^k(Y = S) = \frac{\det(L_S)}{\sum_{|S'|=k} \det(L_{S'})} \propto \det(L_S) = (k! \cdot (\text{Vol}(\text{Conv}(\bar{0} \cup S)))^2), \qquad (12)$$

where $L = B^\top B$, $\text{Conv}(\cdot)$ denotes the convex hull, and $\text{Vol}(\cdot)$ is the $k$-dimensional volume of such a convex hull.

## C.2 Modeling Our Case as A Conditional $N_q^{mb}$-DPP

In this work, we aim to find the compressed subset $D_q^{mb}$ that can best represent the training dataset $D$ by choosing an appropriate compression quality $q$ from its range. To achieve this goal, we introduce $\mathcal{P}_q(D_q^{mb}|D)$ to characterize the conditional likelihood of selecting $D_q^{mb}$ given input $D$ under parameter $q$. Since the network parameter $\theta$ is fixed during compression, and the feature matrix $M_q^c = F_\theta^e(D_q^{mb})$, we rewrite $\mathcal{P}_q(D_q^{mb}|D)$ as $\mathcal{P}_q(M_q^c|D)$ equivalently (see Fig. 11 for the details of constructing $D_q^{mb}$ and $M_q^c$). Depending on the nice properties of DPPs in Sec. C.1, we formulate our goal as a conditional DPP, where $D$ is the associated ground set, and $M_q^c$ is a desired subset in feature space. Then $\mathcal{P}_q(M_q^c|D)$ is a distribution over all subsets of $D$ with cardinality $N_q^{mb}$, which is formally called a conditional $N_q^{mb}$-DPP, since $|M_q^c| = N_q^{mb}$ for any possible subset. Thus, such a DPP formulates the conditional probability $\mathcal{P}_q(M_q^c|D)$ as

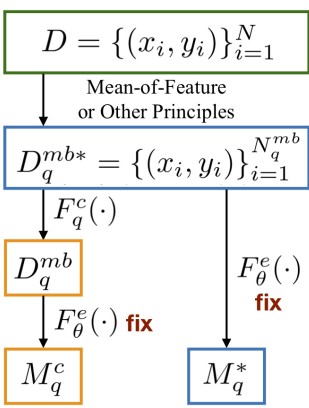

Figure 11: Construction of $D_q^{mb*}$, $D_q^{mb}$, $M_q^*$ and $M_q^c$. $\theta$ is fixed in this stage.

$$\mathcal{P}_q(M_q^c|D) = \frac{\det(L_{M_q^c}(D; q, \theta))}{\sum_{|M|=N_q^{mb}} \det(L_M(D; q, \theta))}, \qquad (13)$$

where $L(D; q, \theta)$ is a conditional DPP $|D| \times |D|$ kernel matrix that depends on the input $D$ parameterized in terms of parameters $\theta$ and $q$. $L_M(D; q, \theta)$ (resp., $L_{M_q^c}(D; q, \theta)$)) is the submatrix sampled from $L(D; q, \theta)$ using indices from $M$ (resp., $M_q^c$).

Similarly, $\mathcal{P}_q(M_q^*|D)$ can be formulated by a conditional $N_q^{mb}$-DPP as

$$\mathcal{P}_q(M_q^*|D) = \frac{\det(L_{M_q^*}(D; \theta))}{\sum_{|M^*|=N_q^{mb}} \det(L_{M^*}(D; \theta))}, \qquad (14)$$

where $|M_q^*| = N_q^{mb}$. Unlike $L(D; q, \theta)$ above, the conditional DPP kernel matrix $L(D; \theta)$ only depends on $D$ and $\theta$ without $q$, since $q$ is fixed at its maximum, i.e., without compression. $L_{M^*}(D; \theta)$ (resp., $L_{M_q^*}(D; \theta)$) is the submatrix sampled from $L(D; \theta)$ using indices from $M^*$ (resp., $M_q^*$).

**Differences:** Of note, two important differences between standard $k$-DPPs and our $N_q^{mb}$-DPP lie in the optimization variable and objective. For this work, we need to find an optimal set cardinality (i.e., $N_q^{mb}$), while it is fixed for standard $k$-DPPs. Although both of them can finally determine a desired subset with the maximum volume, our $N_q^{mb}$-DPP can uniquely determine it by $N_q^{mb}$, while standard $k$-DPPs obtain it by maximizing Eq. (11).

## C.3 THE FIRST GOAL

First, we need to maximize

$$
\begin{aligned}
\mathcal{L}_1(q) = \mathcal{P}_q(M_q^*|D) &= \frac{\det(L_{M_q^*}(D;\theta))}{\sum_{|M^*|=N_q^{mb}} \det(L_{M^*}(D;\theta))} \\
&\propto \det(M_q^{*\top} M_q^*) = (\mathrm{Vol}_q^*)^2 \\
&= (N_q^{mb}! \cdot (\mathrm{Vol}(\mathrm{Conv}(\bar{0} \cup M_q^*)))^2),
\end{aligned}
\tag{15}
$$

where $|M_q^*| = N_q^{mb}$. $L(D;\theta)$ is a conditional DPP $|D| \times |D|$ kernel matrix that depends on the input $D$ parameterized in terms of $\theta$. $L_{M^*}(D;\theta)$ (resp., $L_{M_q^*}(D;\theta)$) is the submatrix sampled from $L(D;\theta)$ using indices from $M^*$ (resp., $M_q^*$). The numerator defines the marginal probability of inclusion for the subset $M_q^*$, and the denominator serves as a normalizer to enforce the sum of $\mathcal{P}_q(M_q^*|D)$ for every possible $M_q^*$ to 1. Here we employ the commonly-used dot product kernel function, where $L_{M_q^*} = M_q^{*\top} M_q^*$ and $L_{M^*} = M^{*\top} M^*$. Conv($\cdot$) denotes the convex hull, and Vol($\cdot$) is the $N_q^{mb}$-dimensional volume of such a convex hull.

Eq. (15) provides a geometric interpretation that the conditional probability $\mathcal{P}_q(M_q^*|D)$ is proportional to the squared volume of $M_q^*$ with respect to $N_q^{mb}$, denoted as $\mathrm{Vol}_q^*$. For our task, $\mathcal{P}_q(M_q^*|D)$ monotonically increases with $N_q^{mb}$. Thus, optimizing $\mathcal{L}_1$ is converted into $\max_q N_q^{mb}$ equivalently (detailed in Proposition 1 as below).

**Proposition 1** *For any matrix $X = (\boldsymbol{x}_1, \boldsymbol{x}_2, \cdots, \boldsymbol{x}_n)$, where $\boldsymbol{x}_i \in \mathbb{R}^d$ is the $i$-th column of $X$, $d > n$ and $X^\top X \neq I$, define $\mathcal{M} = \{i_1, i_2, \cdots, i_m\}$ as a subset of $[n]$ containing $m$ elements, and $X_\mathcal{M}$ as the submatrix sampled from $X$ using indices from $\mathcal{M}$. Then for*

$$
P(\mathcal{M}) := \frac{\det\left(X_\mathcal{M}^\top X_\mathcal{M}\right)}{\sum_{|\mathcal{M}'|=m} \det\left(X_{\mathcal{M}'}^\top X_{\mathcal{M}'}\right)},
$$

*we can ensure that the following two optimization programs*

$$
\max_m P(\mathcal{M})
$$

*and*

$$
\max_m |\mathcal{M}|
$$

*are equivalent.*

Below is one critical lemma for the proof of Proposition 1.

**Lemma 1** *For any matrix $X = (\boldsymbol{x}_1, \boldsymbol{x}_2, \cdots, \boldsymbol{x}_n)$, where $\boldsymbol{x}_i \in \mathbb{R}^d$ is the $i$-th column of $X$ and $d > n$, define $\mathcal{M} = \{i_1, i_2, \cdots, i_m\}$ as a subset of $[n]$ containing $m$ elements, and $X_\mathcal{M}$ as the submatrix sampled from $X$ using indices from $\mathcal{M}$. Then for*

$$
P(\mathcal{M}) := \frac{\det\left(X_\mathcal{M}^\top X_\mathcal{M}\right)}{\sum_{|\mathcal{M}'|=m} \det\left(X_{\mathcal{M}'}^\top X_{\mathcal{M}'}\right)},
$$

*we can ensure that*

$$
[n] = argmax_\mathcal{M} P(\mathcal{M}).
$$

*This means when $|\mathcal{M}| = n$, the probability $P(\mathcal{M})$ takes the maximum value.*

*Proof.* From the definition, we have

$$
\begin{aligned}
0 \leq P(\mathcal{M}) &= \frac{\det\left(X_\mathcal{M}^\top X_\mathcal{M}\right)}{\sum_{|\mathcal{M}'|=m} \det\left(X_{\mathcal{M}'}^\top X_{\mathcal{M}'}\right)} \\
&= \frac{\det\left(X_\mathcal{M}^\top X_\mathcal{M}\right)}{\det\left(X_\mathcal{M}^\top X_\mathcal{M}\right) + \sum_{\{|\mathcal{M}'|=m\} \cap \{\mathcal{M}' \neq \mathcal{M}\}} \det\left(X_{\mathcal{M}'}^\top X_{\mathcal{M}'}\right)} \leq 1.
\end{aligned}
$$

If $|\mathcal{M}'| = m < n$, we have $\binom{n}{m} > 1$ possible options for subset $X_{\mathcal{M}'}$, where $\det\left(X_{\mathcal{M}'}^{\top} X_{\mathcal{M}'}\right) \geq 0$. Only if $|\mathcal{M}| = n$, then $\binom{n}{m} = 1$ and the probability $P(\mathcal{M})$ will definitely take the maximum value, i.e. $P(\mathcal{M}) = 1$. In addition, $X^{\top}X$ has full rank due to $d > n$, and then $\det(X^{\top}X) > 0$ always holds. That is, $P(\mathcal{M}) = 1$ can be guaranteed only when selecting all $n$ items together. This means $[n] = \mathrm{argmax}_{\mathcal{M}} P(\mathcal{M})$. It completes the proof.

**Proof of Proposition 1** From the definition in Deshpande & Rademacher (2010), we have

$$
\begin{aligned}
P(\mathcal{M}) &= \frac{\det\left(X_{\mathcal{M}}^{\top} X_{\mathcal{M}}\right)}{\sum_{|\mathcal{M}'|=m} \det\left(X_{\mathcal{M}'}^{\top} X_{\mathcal{M}'}\right)} \\
&\propto \det\left(X_{\mathcal{M}}^{\top} X_{\mathcal{M}}\right) \\
&= (m! \cdot (\mathrm{Vol}(\mathrm{Conv}(\bar{0} \cup X_{\mathcal{M}})))^2),
\end{aligned}
\tag{16}
$$

where $\mathrm{Conv}(\cdot)$ denotes the convex hull, and $\mathrm{Vol}(\cdot)$ is the $m$-dimensional volume of such a convex hull. Here, from the monotonicity side, it can be easily verified that for $X_{\mathcal{M}} \cup \boldsymbol{x}_{i_{m+1}}$, where $\boldsymbol{x}_{i_{m+1}} \in \complement_X X_{\mathcal{M}}$, we have

$$
(m! \cdot (\mathrm{Vol}(\mathrm{Conv}(\bar{0} \cup X_{\mathcal{M}}^{*})))^2) < ((m+1)! \cdot (\mathrm{Vol}(\mathrm{Conv}(\bar{0} \cup X_{\mathcal{M}} \cup x_{i_{m+1}})))^2),
$$

which means $P(\mathcal{M})$ is monotonically increasing with the increase of $|\mathcal{M}|$.

From the extremum side, by using Lemma 1, we can ensure that the following two optimization programs

$$
\max_{m} P(\mathcal{M})
$$

and

$$
\max_{m} |\mathcal{M}|
$$

have the same optimal value. Thus, maximizing $P(\mathcal{M})$ with respect to $|\mathcal{M}|$ can be converted into $\max_{m} |\mathcal{M}|$ equivalently.

## C.4 THE SECOND GOAL

Second, we need to minimize

$$
\begin{aligned}
\mathcal{L}_2(q) &= \left| \frac{\mathcal{P}_q(M_q^c|D)}{\mathcal{P}_q(M_q^*|D)} - 1 \right| = \left| \frac{\det(L_{M_q^c}(D;q,\theta))}{\det(L_{M_q^*}(D;\theta))} Z_q - 1 \right| \\
&= \left| \frac{\det(M_q^{c\top} M_q^c)}{\det(M_q^{*\top} M_q^*)} Z_q - 1 \right| = \left| \left(\frac{\mathrm{Vol}_q^c}{\mathrm{Vol}_q^*}\right)^2 Z_q - 1 \right| = \left| R_q^2 Z_q - 1 \right|,
\end{aligned}
\tag{17}
$$

where $Z_q = \frac{\sum_{|M^*|=N_q^{mb}} \det(L_{M^*}(D;\theta))}{\sum_{|M|=N_q^{mb}} \det(L_M(D;q,\theta))}$. In particular, $\det(M_q^{*\top} M_q^*)$ has a geometric interpretation that it is equal to the squared volume spanned by $M_q^*$ (Kulesza & Taskar, 2012), denoted as $\mathrm{Vol}_q^*$, likewise for $\det(M_q^{c\top} M_q^c)$ with respect to $\mathrm{Vol}_q^c$. Then we define $R_q = \frac{\mathrm{Vol}_q^c}{\mathrm{Vol}_q^*}$ as the ratio of the two feature volumes. To avoid computing $Z_q$, we can convert optimizing $\mathcal{L}_2$ into minimizing $|R_q - 1|$ equivalently, since both of them mean maximizing $q$ (detailed in Proposition 2 as below).

Below we are going to introduce Proposition 2. We first introduce some necessary assumptions, then we present the theoretical guarantee of Proposition 2.

**Assumption 1** *We make the following assumptions:*

1. ***Image dataset:*** *For any image dataset $D = (x_1, x_2, \cdots, x_n)$, where $x_i$ is the $i$-th image in $D$, define $D_q = (x_1, x_2, \cdots, x_n)$ as a compressed image dataset with the compression quality $q$ for $D$. The compression quality is bounded as $q \in Q$ ($Q$ can be a finite/infinite set), in which the higher of $q$, the better of image quality.*

2. **Feature matrix:** Denote $X = (\boldsymbol{f}_1, \boldsymbol{f}_2, \cdots, \boldsymbol{f}_n) \in \mathcal{R}^{d \times n}$ and $X_q = (\boldsymbol{f}_1^q, \boldsymbol{f}_2^q, \cdots, \boldsymbol{f}_n^q) \in \mathcal{R}^{d \times n}$ as the feature matrices of $D$ and $D_q$ with the same fixed feature extractor, respectively. Here, $d > n$, $X^\top X \neq I$ and $X_q^\top X_q \neq I$.

3. **Selection principle:** Define $\mathcal{M} = \{i_1, i_2, \cdots, i_m\}$ as a subset of $[n]$ containing $m$ elements, and $X_{\mathcal{M}}$ as the submatrix sampled from $X$ using indices from $\mathcal{M}$. Likewise, define $\mathcal{M}_q = \{j_1, j_2, \cdots, j_m\}$ as a subset of $[n]$ containing $m$ elements, and $X_{\mathcal{M}_q}$ as the submatrix sampled from $X_q$ using indices from $\mathcal{M}_q$. Both of them use the same selection principle.

4. **Buffer storage space:** $g(\cdot) : \mathbb{R} \to \mathbb{N}$ represents the function that outputs the maximum number (i.e. $m$ ) such that the compressed version of $D_q$ to a quality $q$ can be stored in the memory buffer. That is, $m$ is uniquely determined by $q$.

**Proposition 2** *Under Assumption 1, for two probabilities*

$$P(\mathcal{M}) := \frac{\det\left(X_{\mathcal{M}}^\top X_{\mathcal{M}}\right)}{\sum_{|\mathcal{M}'|=m} \det\left(X_{\mathcal{M}'}^\top X_{\mathcal{M}'}\right)}$$

*and*

$$P(\mathcal{M}_q) := \frac{\det\left(X_{M_q}^\top X_{M_q}\right)}{\sum_{|\widetilde{\mathcal{M}}|=m} \det\left(X_{\widetilde{\mathcal{M}}}^\top X_{\widetilde{\mathcal{M}}}\right)},$$

*we can ensure that the following two optimization programs*

$$\min_q \left| \frac{\det\left(X_{\mathcal{M}_q}^\top X_{\mathcal{M}_q}\right)}{\det\left(X_{\mathcal{M}}^\top X_{\mathcal{M}}\right)} - 1 \right|$$

*and*

$$\min_q \left| \frac{P(\mathcal{M}_q)}{P(\mathcal{M})} - 1 \right|$$

*are equivalent.*

Below are two critical lemmas for the proof of Proposition 2.

**Lemma 2** *Under Assumption 1, for a ratio*

$$R_q := \frac{\det\left(X_{\mathcal{M}_q}^\top X_{\mathcal{M}_q}\right)}{\det\left(X_{\mathcal{M}}^\top X_{\mathcal{M}}\right)},$$

*we can ensure that*

$$\sup Q = argmin_q |R_q - 1|.$$

*This means when $q = \sup Q$ (i.e, without image compression), $|R_q - 1|$ takes the minimum value.*

*Proof.* From the definition, we know $|R_q - 1| \geq 0$. To minimize $|R_q - 1|$, we need the ratio $R_q = 1$, i.e.,

$$\det\left(X_{\mathcal{M}_q}^\top X_{\mathcal{M}_q}\right) = \det\left(X_{\mathcal{M}}^\top X_{\mathcal{M}}\right). \tag{18}$$

Of note, both $X_{\mathcal{M}_q}$ and $X_{\mathcal{M}}$ are selected from the source dataset $D$ with the same feature extractor, selection principle and set cardinality (i.e., $m$). That is, the only difference between them is image compression with respect to $q$.

By recurring to the Geometric Interpretation in $k$-DPPs (i.e., Eq. (12)), we have

$$\det\left(X_{\mathcal{M}_q}^\top X_{\mathcal{M}_q}\right) = (m! \cdot (\text{Vol}(\text{Conv}(\bar{0} \cup X_{\mathcal{M}_q})))^2)$$

and

$$\det \left( X_{\mathcal{M}}^{\top} X_{\mathcal{M}} \right) = \left( m! \cdot (\mathrm{Vol}(\mathrm{Conv}(\bar{0} \cup X_{\mathcal{M}})))^2 \right).$$

Then, Eq. (18) holds only when $X_{\mathcal{M}_q}$ and $X_{\mathcal{M}}$ have the same convex hull. Due to great difficulty of analytically defining compression function, the two convex hull cannot be mathematically given. However, by conducting extensive experiments on this task (as shown in Fig. 3, Fig. 5, Fig. 8, Fig. 13 and Fig. 14), we find that the volume of $X_{\mathcal{M}_q}$ is larger than that of $X_{\mathcal{M}}$ for a specific $m$ (i.e., $q$). Additionally, with $q$ decreasing (i.e., $m$ increasing), the volume of $X_{\mathcal{M}_q}$ is increasing more quickly. Thus, we have an empirical conclusion that the two volumes are the same only when $q = \sup Q$ (without image compression). It is reasonable since without image compression, the two selection problems about $X_{\mathcal{M}_q}$ and $X_{\mathcal{M}}$ are identical. This means when $q = \sup Q$, $|R_q - 1|$ takes the minimum value, which completes the proof.

**Lemma 3** *Under Assumption 1, for*

$$P(\mathcal{M}) := \frac{\det \left( X_{\mathcal{M}}^{\top} X_{\mathcal{M}} \right)}{\sum_{|\mathcal{M}'|=m} \det \left( X_{\mathcal{M}'}^{\top} X_{\mathcal{M}'} \right)}$$

*and*

$$P(\mathcal{M}_q) := \frac{\det \left( X_{M_q}^{\top} X_{M_q} \right)}{\sum_{|\widetilde{\mathcal{M}}|=m} \det \left( X_{\widetilde{\mathcal{M}}}^{\top} X_{\widetilde{\mathcal{M}}} \right)},$$

*we can ensure that*

$$\sup Q = argmin_q \left| \frac{P(\mathcal{M}_q)}{P(\mathcal{M})} - 1 \right|.$$

*This means when $q = \sup Q$ (i.e, without image compression), $\left| \frac{P(\mathcal{M}_q)}{P(\mathcal{M})} - 1 \right|$ takes the minimum value.*

*Proof.* From the definition, we know $\left| \frac{P(\mathcal{M}_q)}{P(\mathcal{M})} - 1 \right| \geq 0$. To minimize $\left| \frac{P(\mathcal{M}_q)}{P(\mathcal{M})} - 1 \right|$, we need the ratio $\frac{P(\mathcal{M}_q)}{P(\mathcal{M})} = 1$. In fact,

$$\frac{P(\mathcal{M}_q)}{P(\mathcal{M})} = \frac{\det \left( X_{M_q}^{\top} X_{M_q} \right)}{\det \left( X_{\mathcal{M}}^{\top} X_{\mathcal{M}} \right)} \cdot \frac{\sum_{|\mathcal{M}'|=m} \det \left( X_{\mathcal{M}'}^{\top} X_{\mathcal{M}'} \right)}{\sum_{|\widetilde{\mathcal{M}}|=m} \det \left( X_{\widetilde{\mathcal{M}}}^{\top} X_{\widetilde{\mathcal{M}}} \right)} = R_q \cdot Z_q, \tag{19}$$

where $R_q = \frac{\det \left( X_{\mathcal{M}_q}^{\top} X_{\mathcal{M}_q} \right)}{\det \left( X_{\mathcal{M}}^{\top} X_{\mathcal{M}} \right)}$ and $Z_q = \frac{\sum_{|\mathcal{M}'|=m} \det \left( X_{\mathcal{M}'}^{\top} X_{\mathcal{M}'} \right)}{\sum_{|\widetilde{\mathcal{M}}|=m} \det \left( X_{\widetilde{\mathcal{M}}}^{\top} X_{\widetilde{\mathcal{M}}} \right)}$.

By using Lemma 2, we know that when $q = \sup Q$, $R_q = 1$ always holds. More importantly, $Z_q = 1$ also holds if $q = \sup Q$, since the numerator and denominator in $Z_q$ will be equal without compression. That is, when $q = \sup Q$, the ratio $\frac{P(\mathcal{M}_q)}{P(\mathcal{M})} = 1$ and $\left| \frac{P(\mathcal{M}_q)}{P(\mathcal{M})} - 1 \right|$ takes the minimum value. This completes the proof.

**Proof of Proposition 2** From Lemma 2 and Lemma 3, we can ensure that under Assumption 1, the following two optimization programs

$$\min_q \left| \frac{\det \left( X_{\mathcal{M}_q}^{\top} X_{\mathcal{M}_q} \right)}{\det \left( X_{\mathcal{M}}^{\top} X_{\mathcal{M}} \right)} - 1 \right|$$

and

$$\min_q \left| \frac{P(\mathcal{M}_q)}{P(\mathcal{M})} - 1 \right|$$

have the same optimal value. Thus, maximizing $\left| \frac{P(\mathcal{M}_q)}{P(\mathcal{M})} - 1 \right|$ with respect to $q$ can be converted into $\min_q |R_q - 1|$ equivalently, where $R_q = \frac{\det \left( X_{\mathcal{M}_q}^{\top} X_{\mathcal{M}_q} \right)}{\det \left( X_{\mathcal{M}}^{\top} X_{\mathcal{M}} \right)}$.

# D    EMPIRICAL ANALYSIS

## D.1    T-SNE VISUALIZATION OF NORMALIZED FEATURES

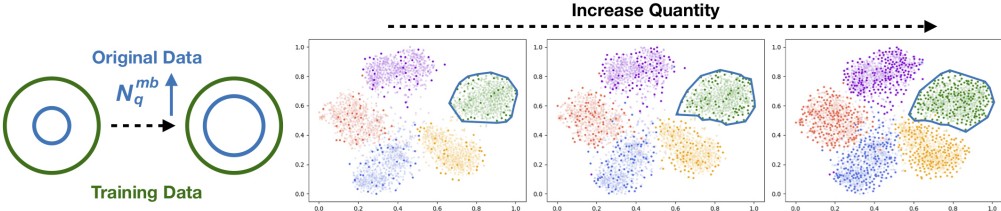

Figure 12: t-SNE visualization of features of original subsets (dark dots) and all training data (light dots) for 5-phase ImageNet-sub with LUCIR. We randomly select five classes out of the latest task and label them in different colors.

To provide an empirical analysis of the quality-quantity trade-off and validate the theoretical interpretation, we use t-SNE (Van der Maaten & Hinton, 2008) to visualize features of all training data, the compressed subset $M_q$, and the original subset $M_q^*$. First, with the increase of quantity $N_q^{mb}$, the area of original subset is expanded and can better cover the training data distribution, as shown in Fig. 12. This result is consistent with maximizing $N_q^{mb}$ for $\mathcal{L}_1$. Second, with the decrease of quality $q$ and increase of quantity $N_q^{mb}$, the compressed data tend to be distorted and thus become out-of-distribution, which has been discussed in the main text Fig. 3. This result is consistent with enforcing $|R_q - 1| < \epsilon$ for $\mathcal{L}_2$.

## D.2    $R_q$ FOR 5-, 10- AND 25-PHASE IMAGENET-SUB

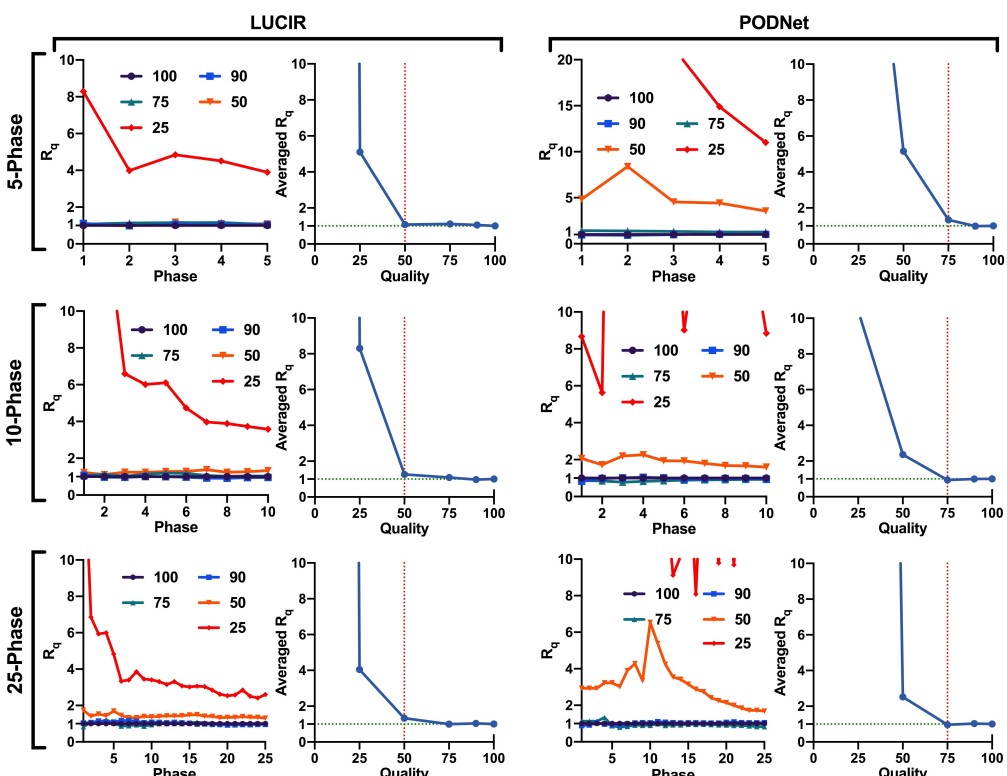

Figure 13: We present $R_q$ in each incremental phase with various compression qualities (left), and the averaged $R_q$ of all incremental phases (right). From top to bottom are 5-, 10- and 25-phase ImageNet-Sub, respectively.

We present $R_q$ in each incremental phase and the averaged $R_q$ of all incremental phases for 5-, 10- and 25-phase ImageNet-sub in Fig.13. Based on the principle in Eq. (5) (we set $\epsilon = 0.5$ as the threshold of $R_q$), it can be clearly seen that 50 and 75 is a good quality for LUCIR and PODNet, respectively. Also, the determined quality is the same for different numbers of splits, which is consistent with the results of grid search in Fig. 2.

## D.3 AANETS AND DDE

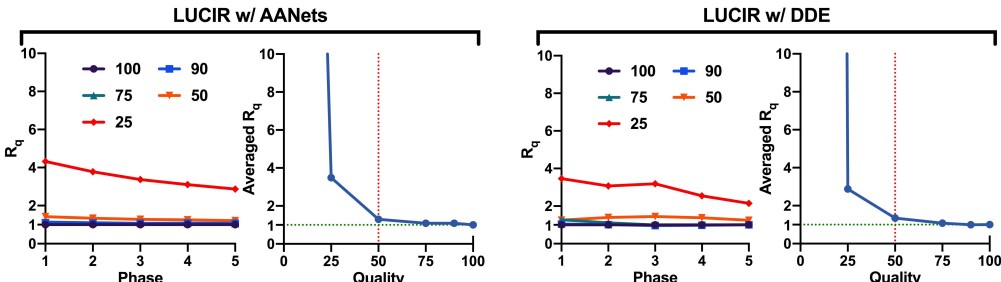

Figure 14: Determine the compression quality for AANets and DDE. We present $R_q$ in each incremental phase (left), and the averaged $R_q$ of all incremental phases (right) for 5-phase ImageNet-sub. The quality of 100 refers to the original data without compression.

Similar to LUCIR and PODNet, we apply the method described in Sec. 4.2 to determine the compression quality for AANets and DDE. Since both AANets and DDE only release their official implementation on LUCIR, here we focus on LUCIR w/ AANets and LUCIR w/ DDE. We present $R_q$ in each incremental phase and the averaged $R_q$ of all incremental phases for 5-phase ImageNet-sub in Fig. 14. The determined qualities for both LUCIR w/ AANets and LUCIR w/ DDE are consistent in different incremental phases, and are the same as that of LUCIR.

## D.4 STATIC VS DYNAMIC QUALITY

For ImageNet-sub with randomly-split classes, whether $|R_q - 1| < \epsilon$ of each quality $q$ is consistent among incremental phases and their average (see Fig. 13 and Fig. 14). Thus, the determined quality by our method usually serves as a static hyperparameter in continual learning, which has been validated by the results of gird search (see Fig. 2). Now, we wonder in which case the determined quality is dynamic, and whether using a dynamic quality is better than using a static one. Intuitively, the determined quality might be affected by the similarity of incremental classes, because the model learned to predict similar classes might be more sensitive to the image distortion caused by data compression.

Since the ImageNet classes are ranked by their semantic similarity, we select 10 adjacent classes as a superclass, and construct 10 adjacent (similar) or remote (dissimilar) superclasses (detailed in Appendix B.1). Similar to the 5-phase ImageNet-sub, we first learn 5 superlcasses as the initial phase, and then incrementally learn one superclass per phase with LUCIR. The results are presented in Fig. 15. For dissimilar superclasses, whether $|R_q - 1| < \epsilon$ is consistent in each incremental phase and their average, so the determined quality is stable at 50, validated by the results of grid search. For similar superclasses, the quality determined by the averaged $R_q$ is 75, where using a stable quality of 75 in-

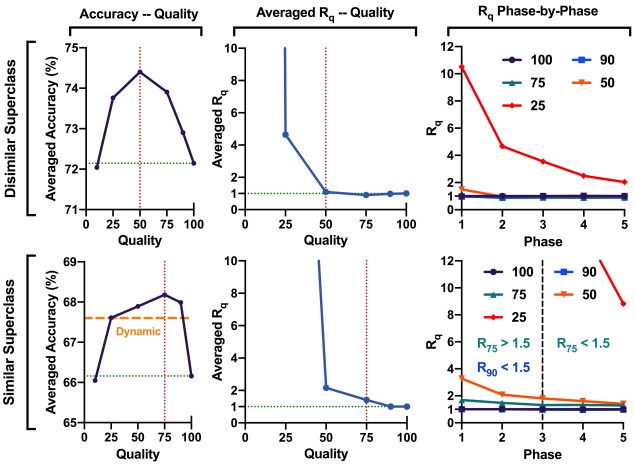

Figure 15: Continual learning of ten similar or dissimilar superclasses with LUCIR.

deed achieves the best performance in grid search. On the other hand, it can be clearly seen that $|R_{75} - 1| > \epsilon$ before phase 3 while $|R_{75} - 1| < \epsilon$ after it. Then we use the dynamically-determined quantity and quantity for each incremental phase. However, the dynamic quality and quantity result in severe data imbalance, so the performance is far lower than using a static quality of either 50, 75 or 90. A promising further work is to alleviate the data imbalance for the scenarios where the determined quality is highly dynamic.

# E    ADDITIONAL RESULTS

## E.1    COMPUTATIONAL COST

In Table 3, we present the detailed results of computational cost and averaged incremental accuracy of LUCIR, LUCIR w/ AANets, LUCIR w/ DDE and LUCIR w/ Ours for CUB-200-2011 and ImageNet-sub.

Table 3: Comparison of computational cost and averaged incremental accuracy. We run each baseline with one Tesla V100.

|  | CUB-200-2011 | | | ImageNet-sub | | |
|---|---|---|---|---|---|---|
| Methods | 5-phase | 10-phase | 25-phase | 5-phase | 10-phase | 25-phase |
| LUCIR | 1.97 h / 44.63% | 3.60 h / 45.58% | 8.03 h / 45.48% | 8.65 h / 70.84% | 10.47 h / 68.32% | 15.63 h / 61.44% |
| LUCIR w/ AANets | 3.52 h / 46.87% | 5.56 h / 47.34% | 14.34 h / 47.35% | 21.94 h / 72.55% | 40.58 h / 69.22% | 91.26 h / 67.60% |
| LUCIR w/ DDE | 7.28 h / 45.86% | 13.09 h / 46.48% | 31.02 h / 46.56% | 55.36 h / 72.34% | 61.81 h / 70.20% | 79.42 h / 66.31% |
| LUCIR w/ *Ours* | 2.70 h / 46.68% | 5.00 h / 47.28% | 11.00 h / 48.01% | 9.09 h / 73.56% | 11.27 h / 72.70% | 18.83 h / 70.53% |

## E.2    STORAGE SPACE

In Table 4, we present the detailed results of different storage space.

Table 4: Averaged incremental accuracy (%) on ImageNet-sub. The storage space of the memory buffer is limited to the equivalent of 10, 20, 40 and 80 original images per class, respectively. The results of LUCIR and PODNet are reproduced from their officially-released codes.

|  | Storage Space | 10 | 20 | 40 | 80 |
|---|---|---|---|---|---|
| 5-phase | LUCIR | 68.83 | 70.90 | 72.64 | 73.56 |
|  | w/ *Ours* | 71.39 / +2.55 | 73.56 / +2.66 | 74.37 / +1.73 | 75.74 / +2.19 |
|  | PODNet | 74.99 | 76.44 | 78.28 | 79.18 |
|  | w/ *Ours* | 76.55 / +1.56 | 78.08 / +1.64 | 79.43 / +1.15 | 80.17 / +0.99 |
| 10-phase | LUCIR | 64.64 | 68.31 | 70.84 | 72.56 |
|  | w/ *Ours* | 68.13 / +3.49 | 72.70 / +4.39 | 74.11 / +3.28 | 76.09 / +3.54 |
|  | PODNet | 70.73 | 73.91 | 76.63 | 78.26 |
|  | w/ *Ours* | 72.78 / +2.05 | 76.02 / +2.11 | 77.82 / +1.19 | 79.27 / +1.01 |
| 25-phase | LUCIR | 58.98 | 64.46 | 67.55 | 70.12 |
|  | w/ *Ours* | 67.75 / +5.77 | 70.53 / +6.07 | 73.49 / +5.94 | 76.03 / +5.91 |
|  | PODNet | 59.41 | 67.17 | 72.57 | 76.21 |
|  | w/ *Ours* | 64.99 / +5.58 | 72.27 / +5.10 | 76.61 / +4.04 | 78.82 / +2.61 |

## E.3    AVERAGED FORGETTING

In addition to averaged incremental accuracy, we evaluate the averaged forgetting, which is calculated by averaging the test accuracy of each task minus its highest test accuracy achieved in continual learning (Caccia et al., 2020). In Table 5, we present the averaged forgetting of classes in the initial phase, which suffers from the most severe forgetting. It can be clearly seen that the averaged forgetting is largely alleviated by ours on each backbone.

Table 5: Averaged forgetting (%) of classes in the initial phase. The storage space is limited to the equivalent of 20 original images per class. DDE (Hu et al., 2021) and AANets (Liu et al., 2021a) are reproduced from their officially-released code.

| | CUB-200-2011 | | | ImageNet-sub | | |
|---|---|---|---|---|---|---|
| Method | 5-phase | 10-phase | 25-phase | 5-phase | 10-phase | 25-phase |
| LUCIR (Hou et al., 2019) | -2.86 | -4.56 | -4.72 | -15.08 | -17.32 | -22.40 |
| w/ *Ours* | -0.84 | -1.08 | -0.75 | -14.11 | -14.64 | -17.96 |
| w/ AANets (Reproduced) | -4.03 | -6.58 | -10.37 | -11.78 | -9.54 | -12.86 |
| w/ AANets + *Ours* | -1.40 | -3.77 | -6.82 | -7.46 | -8.13 | -12.32 |
| w/ DDE (Reproduced) | -1.41 | -5.24 | -3.91 | -14.02 | -17.68 | -23.73 |
| w/ DDE + *Ours* | -0.22 | -0.97 | -0.71 | -11.24 | -12.36 | -17.80 |

## E.4 FIXED MEMORY BUDGET

Here we evaluate memory replay with data compression (ours) under a fixed memory budget (i.e., storage space) on ImageNet-sub. Following a widely-used setting (Rebuffi et al., 2017; Wu et al., 2019), the memory budget (i.e., storage space) is limited to equivalent of 2000 original images (20 original images per class $\times$ a total of 100 classes in ImageNet-sub). We further evaluate a much smaller memory budget, fixed to 1000 original images. Under such a fixed memory budget, ours substantially boosts the performance of each backbone approach as shown in Table 6.

Table 6: Averaged incremental accuracy (%) on ImageNet-sub, under a fixed memory budget of 1000 or 2000 original images.

| | 1000 Original Images | | | 2000 Original Images | | |
|---|---|---|---|---|---|---|
| Method | 5-phase | 10-phase | 25-phase | 5-phase | 10-phase | 25-phase |
| LUCIR (Hou et al., 2019) | 70.16 | 66.32 | 62.07 | 71.87 | 69.37 | 65.72 |
| w/ *Ours* | 72.37 | 69.63 | 68.16 | 73.73 | 73.07 | 72.02 |
| w/ DDE (Reproduced) | 72.61 | 70.82 | 65.58 | 73.47 | 71.92 | 66.70 |
| w/ DDE + *Ours* | 74.40 | 72.62 | 69.03 | 75.35 | 73.93 | 71.17 |

## E.5 LESS COMPRESSED SAMPLES

In Table 7, we present the results of LUCIR with different numbers of compressed samples on ImageNet-sub. Similar to the main text, we select the JPEG quality of 50 for LUCIR, where the storage space of 20 original images can save 85 such compressed images. Memory replay of 85 compressed images of quality 50 achieves a much better performance than that of 20 original images. When reducing the quantity from 85 to 70, 55 and 40, the accuracy will also decline. However, memory replay of 40 compressed images of quality 50 still outperforms that of 20 original images, where the average memory can be saved by 52.94%.

Table 7: Averaged incremental accuracy (%) of LUCIR on ImageNet-sub with different numbers of compressed data. "5-phase", "10-phase" and "25-phase" refer to the accuracy of 5-, 10- and 25-phase ImageNet-sub, respectively.

| Quality | 50 | 50 | 50 | 50 | Original |
|---|---|---|---|---|---|
| Quantity | 85 | 70 | 55 | 40 | 20 |
| Total Storage | 100% | 82.35% | 64.71% | 47.06% | 100% |
| 5-phase | 73.63 | 73.22 | 72.79 | 72.29 | 72.06 |
| 10-phase | 72.65 | 72.01 | 71.35 | 70.16 | 68.59 |
| 25-phase | 70.38 | 68.37 | 66.90 | 65.33 | 63.27 |

