# OpenReview forum: "Memory Replay with Data Compression for Continual Learning"
_ICLR.cc/2022/Conference — ICLR 2022 Poster_

### Official Review · Reviewer_gA5x · 2021-11-01

**Correctness:** 3
**Technical Novelty And Significance:** 4
**Empirical Novelty And Significance:** 3
**Recommendation:** 6
**Confidence:** 4

**Main Review:**

Strengths:

1. The paper is well motivated. The idea is simple but makes sense in the context of memory based continual learning.

2. The experiment to show the tradeoff between the quality and quantity of compressed data is also interesting. It might be expected but the authors conduct a detailed analysis.

3. The experiments are also extensive which show the benefits of the proposed approach.

Weaknesses:

1. The motivation of the proposed method is not that clear. Why maximize the conditional likelihood is a good way to select q?

2. In continual learning, we would like the samples in the memory to represent the old data distribution,  why the proposed objective based on DPP can achieve this?


More questions:


1. Also, Figure 3 is a little confusing, which one is dark dot and which is light dot?

2. Why  keeping the feature volume is a good criteria for selecting q?

3. Is it possible to store examples with mixed quality?

**Summary Of The Paper:**

In this paper, the authors propose a data compression method for memory-replay based continual learning algorithms. With the data compression method,  more old training samples can be stored in the memory to better capture old data distribution. However, there is a trade-off between the quality and quantity of compressed data, the authors propose to use determinantal point processes to determine the quality of the data compression.  Extensive experiments show that with the proposed method, a naive compression method can achieve the SOTA on several continual learning benchmarks.

**Summary Of The Review:**

In this paper, the authors propose to compress the data for memory-based continual learning. Since there is a tradeoff between quantity and quality of the compressed data, the authors propose a novel method to decide the quality of the data. The idea is well-motivated and novel for continual learning.

---

> ### Author Response · Authors · 2021-11-13
> **Author Response to Reviewer gA5x**
>
> Thank you for your valuable comments.
>
> **Q1: More detailed explanation of our method:**
>
> (1) Why is maximizing the conditional likelihood a good way to select q?
>
> (2) Why the proposed objective based on DPP can make the memory represent the old data distribution?
>
> (3) Why is keeping the feature volume a good criteria for selecting q?
>
> A1:
>
> (1) Our learning problem is as follows:
> Assume we receive $T$ training tasks $(D_{t},D_{t}^{mb}), t=1,2,...,T$, where $D_{t}$ and $D_{t}^{mb}$ contain $N_t$ and $N_t^{mb}$ data-label pairs, respectively. In particular, $D_{t}^{mb}$ can best represent the $t$-th training task $D_{t}$ by compressing $N_t^{mb}$ samples in $D_{t}$ to a specific quality.
> For ease of notation, going forward we will assume that the training set contains only a single task $(D,D^{mb})$, and drop the index $t$. All of the following results extend easily to multiple training tasks.
> We let $\mathcal{P}_{q} (D^{mb}|D)$ denote the conditional probability of an output $D^{mb}$ given input $D$ under parameter $q$ (i.e., compression quality).
> The goal of learning is to choose an appropriate $q$ based on the training tasks so that we can make accurate predictions on unseen inputs.
>
> While there are a variety of objective functions commonly used for learning, here we focus on the widely-used maximum likelihood estimation (often abbreviated MLE), where the goal is to choose $q$ to maximize the conditional likelihood of the observed data: $\mathcal{L}(q)=\mathcal{P}_{q} (D^{mb}|D)$.
>
> (2) The construction of $D^{mb}$ can be essentially viewed as a sampling problem with the cardinality ${N}^{mb}$. Here, we apply DPPs to formulate the conditional likelihood/probability objective function $\mathcal{P}_{q} (D^{mb}|D)$, because DPPs are not only elegant probabilistic sampling models, which can characterize the probabilities for every possible subset by determinants, but also provide a geometric interpretation of the probability by the volume spanned by all elements in the subset.
>
> In particular, a conditional DPP is a conditional probabilistic model which assigns a probability $\mathcal{P_{q}} (D^{mb}|D)=\frac{\det(L_{D^{mb}}(D; q)}{\sum_{|{D'}|=N^{mb}}\det(L_{{D'}}(D; q))}$ to each possible subset $D^{mb}$.
> Optimizing $\mathcal{L}(q)$ is consistent under mild assumptions; that is, if the training data are actually drawn from a conditional DPP with parameter $q$, then the learned $q \rightarrow q^*$ as $T \rightarrow \infty$.
> Of course real data are unlikely to exactly follow any particular model, but in any case the maximum likelihood approach has the advantage of calibrating the DPP to produce reasonable probability estimates as claimed in [2*], since maximizing $\mathcal{L}(q)$ can be seen as minimizing the loss on the training data.
>
> (3) Our theoretical analysis based on DPPs makes keeping feature volume of  $D^{mb}$ (i.e., the compressed subset) similar to that of $D^{mb*}$ (i.e., the original subset) a good criteria to select $q$.
> Specifically, for efficient computation, our objective function $\mathcal{P_{q}} (D^{mb}|D)$ is converted into two goals: to maximize $\mathcal{P_{q}} (D^{mb}|D)$, we can maximize $\mathcal{P_{q}} (D^{mb*}|D)$ instead; and meanwhile to constrain that $\mathcal{P_{q}} (D^{mb}|D)$ is consistent with $\mathcal{P_{q}} (D^{mb*}|D)$. Based on the properties of DPPs, the probability $\mathcal{P_{q}} (D^{mb}|D)$ (reps., $\mathcal{P_{q}} (D^{mb*}|D)$) is related to the volume spanned by its associated feature vectors.
> So, such a constraint further corresponds to keeping the feature volume of $D^{mb}$ similar to that of $D^*$. (Please refer to Appendix C for the complete proof.)
>
> Our empirical results in Fig. 3 are indeed consistent with our theoretical modelling based on DPPs, where we find that the best performance is achieved when the feature area/volume of compressed data is similar to that of original data.
>
> [2*] Alex Kulesza and Ben Taskar. Determinantal point processes for machine learning. arXivpreprint arXiv:1207.6083, 2012.
>
>
> **Q2: Clarity of Figure 3:**
>
> A2: Thanks. In Figure 3, we use t-SNE to visualize the features of the original subset $M_q^*$ (light dot), which includes different amounts of original data, and its compressed subset $M_q^c$ (dark dot), which is obtained by compressing the original subset to just fit in the memory buffer. Each color represents a class. From left to right, the amount of data included in a subset is 37, 85 and 200, while the JPEG quality of the compressed subset is 90, 50 and 10, respectively. We will make it clearer in the final version.

---

> > ### Author Response · Authors · 2021-11-13
> > **Author Response to Reviewer gA5x Q3**
> >
> > **Q3: Is it possible to store examples with mixed quality?**
> >
> > A3: Yes. Indeed, the compression quality for each incremental data can be dynamically determined by our method described in Section 4.2, through calculating the feature volume ratio $R_q$ among several candidate qualities.
> > As shown in Fig. 5, we can calculate $R_q$ for the training data in each incremental phase. Interestingly, whether $|R_q − 1| < \epsilon$ for each $q$ is generally consistent in each incremental phase for both class-incremental learning in Section 5.1 (see Fig. 5, Appendix D.2 and D.3) and semi-supervised continual learning of object detection in Section 5.2 (see Fig. 8). Thus, for the above two scenarios, using a stable compression quality determined by averaging $R_q$ of all incremental phases is actually the same as dynamically determining it for each incremental phase.
> >
> > In Appendix D.4, we further explore in which case the determined quality might be more dynamic, and whether using a dynamic quality is better than using a static one. We observe that the determined quality will be more dynamic if the incremental classes are more similar, since the decision boundary is more sensitive to the distortion caused by data compression. However, different qualities enable the memory buffer to save different amounts of compressed data for each class, resulting in severe data imbalance (detailed in Appendix D.4, Fig. 15).
> >
> > As a simple but effective solution, we recommend to use a fixed quality determined by averaging $R_q$ of several incremental phases (if applicable), or even by $R_q$ of the first phase. This strategy can indeed achieve a considerable performance (see Appendix D.4, Fig. 15). As numerous efforts have been devoted to addressing data imbalance in memory replay, such as by cosine normalization (LUCIR, CVPR2019), bias correction layer (BiC, CVPR2019) or removing the biased momentum effect in SGD (DDE, CVPR2021), we leave it as a future extension to alleviate the data imbalance for the scenarios where the determined quality is highly dynamic.

---

### Official Review · Reviewer_iR3u · 2021-11-02

**Correctness:** 4
**Technical Novelty And Significance:** 1
**Empirical Novelty And Significance:** 2
**Recommendation:** 3
**Confidence:** 4

**Main Review:**

Strengths
1. The problem of catastrophic forgetting has gained considerable attention.

2. The paper is written well and easy to follow.

3. Experiments support the claims.

Weaknesses:

1. The idea is not novel.

2. Unnecessary complexities in the implementation of the idea.

3. Analytical and ablative experiments are limited.

**Summary Of The Paper:**

In this paper, the problem of catastrophic forgetting for continual learning is explored. The core idea is to benefit from data compression to reduce the space to store data samples and then replay the reconstructed data points that are built using the compressed versions. JPEC method has been explored empirically to determine an optimal compression rate through solving an optimization problem. Experiments on three benchmark datasets are performed to demonstrate that the method is effective.

**Summary Of The Review:**

1. The proposed idea is not novel. It is somewhat trivial that whenever we have storage limitations, data compression is going to be helpful. This on its own is not an idea at the level of a top-tier venue.


2. I find subsection 4 to be an unnecessary section to make the idea look non-trivial but I think it only is complicating the method unnecessarily.  A naive grid search can do the job. The justification that the authors have provided is that grid search is of "huge computational cost". But first of all, it is not clear how large this computational cost is. Second, even if it is large, it is not going to be a huge burden for continual learning. Because you can determine q one time before starting model execution. This is what we do in most cases when we want to tune a hyperparameter. There is no theoretical contribution in that section either and along with the appendices, only known results are rewritten. As a result, the idea contribution of this manuscript is highly limited.


3. In experiments, it is mentioned that 20 samples per class are stored. However, this is going to lead to a memory buffer that is going to grow as more classes are used. The appropriate way to do this is to consider a memory buffer with a fixed size and then discard samples as new tasks are learned to replace a portion of old samples with new samples.

4. In the results, standard deviation should be added to make the comparison more informative.

5. It is not very clear how much advantageous data compression would be for continual learning. An additional analytic experiment can be to show that how much memory can be saved by storing less compressed samples while getting no considerable performance degradation.

---

> ### Author Response · Authors · 2021-11-13
> **Author Response to Reviewer iR3u**
>
> Thank you for your valuable comments. Below we provide point-to-point response to the comments, especially to the novelty and necessity of our proposal.
>
> **Q1: Novelty and the idea is trivial**
>
> A1: We appreciate the reviewer for the comment but disagree with our highest respect. When we have storage limitations, data compression is  **not necessarily helpful** for memory replay. In fact, with an improperly selected compression quality $q$, the performance might be even worse than that without compression, as demonstrated in Section 4.1 Fig. 2. Therefore, we argue that to determine a proper quality-quantity trade-off is necessary. Further, simplicity and novelty are not contradictory. In fact, as agreed by Reviewer gA5x and Reviewer iLt4, ''The idea is well-motivated and novel for continual learning'', and ''It is generally clear, well-written and insightful''.
>
> **Q2: Necessity of Section 4:**
>
> A2: Thanks for the comment and we clarify the necessity of the proposal in Section 4. In particular, comparing to the naive grid search, our proposal requires **$100\times$ less computation time** to determine a proper compression quality **without accessing to future data or revisiting the learned data**, as presented below in detail.
>
> (1) Computation cost:
>
> Using the naive grid search, determining $q$ results in a huge extra computation cost, since it needs to additionally learn the task sequence for several times. For example, training LUCIR with memory replay of 20 images per class on 5-phase ImageNet-sub needs $8.65$ hours on one Tesla V100 (see Fig. 6). In this case, the naive grid search of five candidate qualities needs $5 \times 8.65  = 43.25$ extra hours. In comparison, the extra computation cost of our proposal is no more than $20$ minutes, which speeds up the grid search approach $100$ times. The efficiency of our proposal is even more valuable in recent approaches such as PODNet (ECCV2020), AANets (CVPR2021) and DDE (CVPR2021), because they are much slower than LUCIR generally.
>
> (2) Applicability in realistic scenario:
>
> The naive grid search of hyperparameters needs to recurrently visit a task sequence (or a sub-sequence), which conflicts with the restriction of continual learning. If the old data cannot be revisited, or the future data cannot be accessed immediately, the naive grid search will be inapplicable to determine the quality. In contrast, ours can determine a proper compression quality for the currently-arrived training data just after learning them, without accessing to future data or revisiting the learned data. Therefore, ours is more applicable to handle such realistic challenges of continual learning.
>
> In conclusion, the development of our proposal is necessary considering the efficiency and natural restrictions of continual learning. We will clarify the necessity of Section 4 in the finial version.
>
>
> **Q3: Memory buffer with a fixed size:**
>
> A3: Thanks for the valuable suggestion. Under a fixed memory budget, memory replay with data compression (ours) can still achieve a considerable performance improvement. In fact, we added an experiment on ImageNet-sub, presented in the Rebuttal Table 2 as detailed below. Following a widely-used setting (iCaRL, CVPR2017; EEIL, ECCV2018; BiC, CVPR2019), the memory budget is equivalent to 2000 original images (20 original images per class $\times$ a total of 100 classes in ImageNet-sub). We further evaluate a much smaller memory budget, fixed to 1000 original images. It can be clearly seen that our proposal substantially boosts the performance of LUCIR and DDE in both the two scenarios. We are still running the experiments in other scenarios and will add all additional results in the final version.
>
>
>
> **Rebuttal Table 2.** Averaged incremental accuracy (\%) of classification tasks on ImageNet-sub, under a fixed memory budget of 1000/2000 original images (refer to as ''budget=1000/2000'').
>
> | Method |budget=1000, 5-phase | budget=1000, 10-phase |budget=1000, 25-phase |budget=2000, 5-phase | budget=2000, 10-phase | budget=2000, 25-phase |
> | :----: | :----: | :----: | :----: | :----: | :----: | :----: |
> |LUCIR                    | 70.16 | 66.32 |62.07 |71.87 |69.37 |65.72 |
> |LUCIR w/ Ours    | 72.37 | 69.63 |68.16 |73.73 |73.07 |72.02 |
> |DDE                        | 72.61| 70.82 |65.58 | 73.47 | 71.92 | 66.70 |
> |DDE w/ Ours        | 74.40| 72.62 |69.03 | 75.35 | 73.93 | 71.17 |
> |  |  | | | | | | | | |
>
>
> **Q4: Standard deviation should be added:**
>
> A4: Thanks for the valuable suggestion. We added the standard deviation and the improvement of the proposal is significant compared to the measurement variance. We will add it in the updated version.

---

> > ### Author Response · Authors · 2021-11-13
> > **Author Response to Reviewer iR3u Q5**
> >
> > **Q5: How much memory can be saved by storing less compressed samples?**
> >
> > A5: Thanks for the valuable suggestion. The performance will decrease as the amount of compressed data decreasing. Nevertheless, when more than half of the storage is saved, our approach is still better than that of the baseline.
> >
> > In Rebuttal Table 3, we present the accuracy of LUCIR on 5-, 10- and 25-phase ImageNet-sub with different numbers of compressed data. As discussed in Section 4 Fig. 2 and Fig. 5, we select the JPEG quality of $50$ for LUCIR, where the storage space of $20$ original images can save $85$ such compressed images. Memory replay of $85$ compressed images of quality $50$ indeed achieves a much better performance than that of $20$ original images. When reducing the quantity from $85$ to $70$, $55$ and $40$, the accuracy will also decline. However, memory replay of $40$ compressed images of quality $50$ still outperforms that of $20$ original images, where the average memory can be saved by 52.94\%.
> >
> > **Rebuttal Table 3.** Averaged incremental accuracy (\%) of LUCIR on ImageNet-sub with different numbers of compressed data. ''5-phase'', ''10-phase'' and ''25-phase'' refer to the averaged incremental accuracy of 5-, 10- and 25-phase ImageNet-sub, respectively.
> >
> > |   |  |  |   |  |  |
> > | :----: | :----: | :----: | :----: | :----: | :----: |
> > |Quality  | 50 | 50 | 50 | 50 | Original |
> > |Quantity| 85 | 70 | 55 | 40 | 20 |
> > |Total Storage|100\%|82.35\%|64.71\%|47.06\%|100\%|
> > |5-phase |73.63 |73.22 |72.79 |72.29 |72.06 |
> > |10-phase |72.65 |72.01 |71.35 |70.16 |68.59 |
> > |25-phase |70.38 |68.37 |66.90 |65.33 |63.27 |
> > |   |  |  |   |  |  |

---

> > > ### Comment · Reviewer_iR3u · 2021-11-17
> > > **Response to the authors**
> > >
> > > Thank you for your diligent works and for providing additional experiments.  I understand novelty is a subjective notion but just using data compression in continual learning in my opinion is not enough to warrant publication in a conference similar to ICLR. All the work around tuning q theoretically in my opinion is just spending energy on something without substantial importance in the field. In conclusion, I still think the novelty of this work is very limited and I don't see enough merit for publication at ICLR. I maintain my score.

---

> > > > ### Author Response · Authors · 2021-11-17
> > > > **Author Response**
> > > >
> > > > We sincerely appreciate the reviewer for the comments but we strongly disagree with our highest respect. As emphasized in the rebuttal, our proposal is not ''just using data compression in continual learning'', but opens up a promising new avenue for continual learning. The key issue in this new avenue is to address the quality-quantity trade-off in a computation-efficient and applicable way, which is the main technical focus of this paper. Therefore, we argue that this paper has its own novelty.

---

### Official Review · Reviewer_kRF2 · 2021-11-02

**Correctness:** 3
**Technical Novelty And Significance:** 2
**Empirical Novelty And Significance:** 2
**Recommendation:** 6
**Confidence:** 4

**Main Review:**

Strengths :

1. This proposed method demonstrates its advantages in realistic continual learning scene.
2. The experiment of this paper is sufficient.
3. The method is easy to follow, and the performance is improved to a certain extent.
4.The paper is well written, and the supplementary material is abundant.
Weaknesses :
1. The novelty of this paper is incremental.
2. Only the average accuracy is reported. There is a lack of some other common used metrics like average forgetting.


**Summary Of The Paper:**

In this work, the authors propose memory replay with data compression, which is both an important yet neglected baseline and a promising direction for continual learning. Using a naive technique of data compression with a properly selected quality, the proposed method can achieve the SOTA performance in a time-efficient and plug-and-play way. Since the compression quality is highly nontrivial for the efficacy of memory replay, the authors provide a novel method based on determinantal point processes (DPPs) to determine it efficiently, and validate their method in both class-incremental learning and semi-supervised continual learning of object detection.

**Summary Of The Review:**

 The novelty of this paper is incremental but the experiment of this paper is sufficient.

---

> ### Author Response · Authors · 2021-11-13
> **Author Response to Reviewer kRF2**
>
> Thank you for your valuable comments.
>
> **Q1: Novelty:**
>
> A1: Although memory replay can generally achieve the best performance in most continual learning scenarios, existing work mainly focuses on more effectively constructing and exploiting a memory buffer of a few original data, which will lose a lot of information about the old data distribution and require a lot of extra efforts (computation) to improve the performance.
>
> The novelty of this paper is two-folded. First of all, we propose data compression as an additional degree of freedom for memory replay. It can easily achieve the SOTA performance with minimal extra computation, and thus opens a promising new avenue for continual learning. Further, we extensively investigate the highly nontrivial issue in this new avenue, i.e., the quality-quantity trade-off, and propose an effective solution to address it in a computation-efficient way. (Please refer to the response to Reviewer iLt4 Q1, and to Reviewer iR3u Q1 and Q2 for details.) We will make it clearer in the final version.
>
> **Q2: Additional metric like average forgetting:**
>
> A2: Thanks for the suggestion. We conduct extensive extra experiments under the metric of averaged forgetting and the results agree with the main claim of the paper. In particular, we evaluate the averaged forgetting for class-incremental learning described in Section 5.1. The extra experiments demonstrate that memory replay with data compression (ours) can effectively mitigate catastrophic forgetting of the old classes, as detailed below:
>
> In Rebuttal Table 1, we present the averaged forgetting of classes in the initial phase, which suffers from the most severe forgetting. It can be clearly seen that the averaged forgetting is largely alleviated by ours on each backbone approach. We are still running the experiments on other baselines and benchmarks, and will add all additional results in the final version.
>
> **Rebuttal Table 1.** Averaged forgetting (\%) of classes in the initial phase. Both AANets and DDE are implemented on the backbone of LUCIR, reproduced from their officially-released code.
>
> |Method | CUB 5-phase | CUB 10-phase | CUB 25-phase | ImageNet-sub 5-phase | ImageNet-sub 10-phase | ImageNet-sub 25-phase |
> | :----: | :----: | :----: | :----: | :----: | :----: | :----: |
> |LUCIR                     |-2.86|-4.56|-4.72|-15.08 |-17.32 |-22.40 |
> |LUCIR w/ Ours    |-0.84 |-1.08|-0.75|-14.11 |-14.64 |-17.96 |
> |AANets                   |-4.03 |-6.58 |-10.37 |-11.78 |-9.54 |-12.86 |
> |AANets w/ Ours   |-1.40|-3.77 |-6.82 |-7.46 |-8.13 |-12.32 |
> |DDE                         |-1.41 |-5.24 |-3.91|-14.02 |-17.68 |-23.73 |
> |DDE w/ Ours        |-0.22|-0.97|-0.71|-11.24 |-12.36 |-17.80 |
> | | | | | | | |

---

> > ### Comment · Reviewer_kRF2 · 2021-12-01
> > **Response to the authors**
> >
> > Thank you for your diligent works and for providing additional experiments. And I think the proposed method is a necessary component for practical applications of compressed data replay. I keep my final decision as "marginally above the acceptance threshold".

---

> > > ### Author Response · Authors · 2021-12-01
> > > **Thanks for the feedback**
> > >
> > > Thank you very much for the positive feedback! We highly appreciate that.

---

### Official Review · Reviewer_iLt4 · 2021-11-03

**Correctness:** 3
**Technical Novelty And Significance:** 3
**Empirical Novelty And Significance:** 3
**Recommendation:** 6
**Confidence:** 4

**Main Review:**

Strengths:
-	The empirical comparisons appear to be performed to good standards.
-	The results are convincing, even if not very surprising.
-	Practical benefit of the proposed memory replay with compression approach is demonstrated on a large scale object detection task.

Weaknesses / suggestions:
-	The claim that memory replay with compression is neglected is somewhat too strong. In particular, I think there are two existing lines of work that would be good to discuss:
* Firstly, there is highly related previous work addressing the problem of online continual compression: http://proceedings.mlr.press/v119/caccia20a.html
* Secondly, one motivation for generative replay (rather than replaying stored samples) is that learning a generative model can result in a compressed representation of the original data (even if in practice this is not always the case).
-	Section 3 is not completely clear and somewhat confusing. Based on the description in this section, I would think that this paper deals with domain-incremental learning (https://arxiv.org/abs/1904.07734) or the new instances setting (http://proceedings.mlr.press/v78/lomonaco17a.html), but from the rest of the paper I understand it deals with class-incremental learning / the new classes setting.
-	The paper generously uses the term state-of-the-art (SOTA). In almost all cases, I don’t think the use is fully justified. I also don’t think the use of this term is relevant or necessary for this paper, and I would recommend removing most if not all mentions of SOTA in the paper.


**Summary Of The Paper:**

This paper studies the problems of classification and object detection from natural image datasets in a continual learning setting, whereby the different classes/objects to be learned are not observed together but sequentially. It is assumed that a memory buffer of a certain size is available to store data in. The strategy used by the paper is to fill this memory buffer with compressed data samples (with JPEG) rather than the original images. Such compression introduces a quantity-quality trade-off, which this paper empirically analyses. Additionally, the paper proposes an automated way to select the amount of data compression based on determinantal point processes.

**Summary Of The Review:**

I think this is a solid paper. It is generally clear, well-written and insightful, even if its contributions are not surprising or technically challenging. I support acceptance.

---

> ### Author Response · Authors · 2021-11-13
> **Author Response to Reviewer iLt4**
>
> Thank you for your valuable comments.
>
> **Q1: Too strong claim and two lines of related work to discuss:**
>
> A1: Thanks for pointing out the related work. We will modify our claim properly and discuss the related work in the finial version. In comparison, our proposal enjoys a computational benefit in terms of running time and can scale up to a realistic setting.
>
> In particular, continual learning needs to address catastrophic forgetting through efficient computation and storage. Compared with ours, a common issue of both the online continual compression [1*] and generative replay is that they need to learn an additional generative model, which is generally computation-inefficient. In addition to saving old data, [1*] also needs to store the generative model, which brings extra storage cost.
>
> Besides, due to the difficulty of continually learning a generative model from online data stream, the application of [1*] and generative replay for continual learning is generally limited to relatively simple domains such as CIFAR-10. (The effect of [1*] on larger images is only evaluated in an "offline iid" training way, rather than continual learning.) In contrast, we focus on addressing continual learning of more challenging but realistic large-scale images with more complex domains, such as ImageNet and even larger SODA10M, which is a common focus of numerous recent work in continual learning.
>
> Further, the motivation of [1*] is almost orthogonal to ours. [1*] attempted to learn a neural compression model from online data stream. In contrast, we extensively analyze how the quality-quantity trade-off in data compression affects memory replay, and then provide a novel method to efficiently determine an appropriate compression quality. Indeed, the combined effect of quality and quantity is an under addressed issue in [1*], as they claimed ''simply having more stored memories does not amount to better performance as their quality may be severely degraded and affected by drift'' when processing large-scale images. Therefore, [1*] as a data compression approach can be naturally adapted to our framework, and our method to determine the compression quality can further boost [1*] particularly on large-scale images.
>
> [1*] Online learned continual compression with adaptive quantization modules. ICML2020.
>
>
> **Q2: Clarity of Section 3:**
>
> A2: Thanks. The main focus in this paper is the class-incremental learning / new classes setting, as studied in Section 3, 4 and 5.1. To improve clarity, we will explicitly interpret our setting as ''to incrementally learn numerous tasks from their task-specific training dataset'', ''For classification tasks, the training samples of each task might be from one or several new classes, i.e., class-incremental learning.'' in the updated version.
>
>
> **Q3: The use of state-of-the-art (SOTA):**
>
> A3: Thanks for your suggestion. However, we double-checked all of the "SOTA" terms used in our paper. All cases divide into two categories:
>
> In the first category, it refers to previously published results. For instance, AANets (CVPR21) and DDE (CVPR21) achieve the best performance in a general setting of class-incremental learning firstly proposed by LUCIR (CVPR19), followed by PODNet (ECCV20), TPCIL (ECCV20) and Mnemonics (CVPR20).
>
> In the second one, it refers to our results, which clearly outperform the existing methods in a fair comparison. For instance, using the same memory buffer size and the same backbone, the performance of ours is better than AANets and DDE (the results are reproduced from their officially-released code). Combined with AANets or DDE, the performance of ours is further greatly improved.
>
> Therefore, we try our best to justify the use of the "SOTA" terms. To improve clarity, we will remove some redundant terms in the updated version.

---

> > ### Comment · Reviewer_iLt4 · 2021-11-16
> > **Reviewer Response to First Author Rebuttal**
> >
> > Thanks for your fast response and for the updates to the paper.
> >
> > Section 3 is improved. However, although I think it is now probably clear from this paragraph what type of continual learning problem the paper deals with, I don’t think this paragraph is correct. In particular, it is not described in this paragraph that the algorithm must learn to distinguish between classes from different tasks (or that the algorithm must also infer task identity). For most people this is probably implied by using the term “class-incremental learning”, but I think it is important to make this explicit. It might be useful to refer to https://arxiv.org/abs/1904.07734.
> >
> > Thank you for your explanation of how the use of the term SOTA is intended, this certainly helps. However, I still think the use of this term in the paper is problematic. Most importantly, I’m not convinced that a direct comparison between the results of this paper with the results of LUCIR (CVPR19), PODNet (ECCV20), TPCIL (ECCV20) and Mnemonics (CVPR20) is fair. I think at least some of those papers defined the problem setting they were interested in as that a certain number of images from previous tasks could be stored. The current paper defines the problem setting they are interested in as that a certain number of bits from previous tasks could be stored. These are different rules. A direct comparison between these, and the use of the term SOTA, might therefore be somewhat misleading.
> > I don’t think the contribution of this paper is proposing a new state-of-the-art method, and I don't think that is a problem as long as it is clearly communicated. I think the contribution of this paper is exploring the quantity/quality trade-off in memory replay.

---

> > > ### Author Response · Authors · 2021-11-17
> > > **We revised our paper and uploaded a new version.**
> > >
> > > Thank you for your suggestions. We properly revised our paper in response to these comments and uploaded a new version.
> > >
> > > **Clarity of Section 3:**
> > >
> > > In Section 3, we further clarify the setting as ''For classification tasks, the training samples of each task might be from one or several new classes. At test time, all the classes ever seen are evaluated, and the classes from different tasks need to be distinguished. This setting is also called class-incremental learning [3*].''
> > >
> > > [3*] Van de Ven, G. M., \& Tolias, A. S. (2019). Three scenarios for continual learning. arXiv preprint arXiv:1904.07734.
> > >
> > > **The use of state-of-the-art (SOTA)**
> > >
> > > Thanks for clarification of your questions.
> > >
> > > We agree that one of the main purposes is to explore the quality quantity trade-off in data comparison for memory replay. However, it is also important to see the effectiveness of data compression by applying it to various baselines. Following your suggestions, we change our claim as ''Ours can largely improve the efficacy of memory replay'', ''Given a limited storage space, ours can achieve comparable or more performance improvements than the recent strong approaches'', and ''Ours can greatly improve the performance of recent strong baselines'', so as to demonstrate the effectiveness of data compression with a proper quantity-quality trade-off.
> > >
> > > We sincerely appreciate the reviewer for the quick response and we believe the clarify of the paper is improved. If you have any further questions, please let us know.

---

> > > > ### Comment · Reviewer_iLt4 · 2021-11-17
> > > > **Reviewer Response to Second Author Rebuttal**
> > > >
> > > > Thank you for your fast response and the further improvements.
> > > >
> > > > I would suggest to slightly modify your claims such as "ours can greatly improve the performance of recent strong baselines" by indicating that this improvement is made possible by viewing the limitation of the memory buffer in terms of bits rather than in terms of images. Another example, in the second paragraph of section 5.1, you say that "the same as representative memory replay approaches (Hou et al., 2019; Douillard et al., 2020)". It would be good to add that a difference is that in your implementation the memory buffer can be filled with a certain number of bits, while in the previous implementations the memory buffer was filled with a certain number of images.
> > > >
> > > > Essentially, I don't think it's necessarily the case that previous work "didn't think of trying to compress the stored images", it might have been that they were simply interested in different constraints.

---

> > > > > ### Author Response · Authors · 2021-11-18
> > > > > **Our paper has been updated.**
> > > > >
> > > > > Thank you for your valuable suggestions and quick response. Following these suggestions, we further clarify that ''ours can greatly improve the performance of recent strong baselines'' is "by saving more compressed data in a limited storage space". In the updated version, we also replace the use of ''buffer size'' by ''storage space'' to make it clearer.
> > > > > Further, in Section 5.1 we clarify the implementation as ''We follow the implementation of representative memory replay approaches (Hou et al., 2019; Douillard et al., 2020) (detailed in Appendix B.1), where we focus on constraining a certain storage space of the memory buffer rather than a certain number of images.''
> > > > >
> > > > > We uploaded a new version that includes the above revision. Please let us know if you have any further question.

---

### Author Response · Authors · 2021-11-13
**Summary of Changes in Currently-Revised Version**

We thank all reviewers for their valuable comments. We are revising our paper in response to these comments, and have submitted a currently-updated version. Here we summarize changes of the currently-updated version as detailed below:

1. Add two lines of related work and modify our claim: We discuss the two lines of related work (i.e., generative replay and the online continual compression) in the third paragraph in Section 2. Also, we modify our claim of the proposed memory replay with data compression as ''an important baseline and a promising direction for continual learning''.

2. Improve clarity of Section 3:  Please refer to the first paragraph in Section 3. For classification tasks, we emphasize that our work mainly focuses on class-incremental learning.

3. Improve clarity of describing Fig. 3: Please refer to the third paragraph in Section 4.2.

4. Further clarify the necessity of determining the compression quality without additional training: We emphasize the advantages of our method described in Section 4.2 in terms of computation-efficiency and applicability, compared with the naive grid search approach. Please refer to the third paragraph in Section 1 and the first paragraph in Section 4.3.

5. Modify our claim of ''SOTA''.

6. Add the standard deviance for the results in Table 1 and Table 2.

7. Add the results of average forgetting in Appendix E.3.

8. Add the results of memory buffer with a fixed size in Appendix E.4.

9. Add the results of storing less compressed samples in Appendix E.5.


We are still improving our paper and will submit a final updated version after finishing all revisions. Please let us know if you have any extra comments and suggestions. We could further improve them in the final updated version.

---

### Author Response · Authors · 2021-11-19
**Looking forward to your reply**

Dear reviewers,

Thanks a lot for your efforts in reviewing this paper. We tried our best to address all mentioned concerns/problems, and properly revised our paper in response to the comments. We hope you might view this as sufficient reason to further raise your score. If your have any further questions, please let us know.

Best,

Authors

---

### Decision · Program_Chairs · 2022-01-20

**Decision:**

Accept (Poster)

**Comment:**

This works considers limitations of rehearsal-based methods in the context of continual learning (classification and object detection). Rehearsal-based methods provide a strong baseline, but a loss in predictive performance arises when the memory is limited in size. The authors propose to leverage compression (JPEG) to increase the number of data (images) stored in the memory. The approach is evaluated in the context of an autonomous driving application.

The additional experiments conducted by the authors were highly appreciated and helped clarify open questions (e.g., class-incremental learning set-up, DPP objective to determine size of the memory, quantity vs quality of compressed data, etc.). The authors addressed the issues raised by three out of four reviewers, who did not have further comments. The remaining reviewer found that the methodological contributions of this paper, namely of using compression in the context of CL, was pretty straightforward. However, the authors addressed the concerns raised by the reviewer regarding the selection of the compression quality q as far as I am concerned and conducted additional experiments to further demonstrate the usefulness of the approach. I would encourage the authors to include this discussion in the final version of the paper. I would also encourage them to include the additional experiments they conducted with fixed memory size and amount of memory that can be saved.